# Seismic Behaviour of CFST Space Intersecting Nodes in an Oblique Mesh

**Jun Zhao** [1,†], **Feicheng Wang** [2,*], **Bai Yang** [1,†] **and Bin Ma** [1,*]

1   School of Architecture and Transportation Engineering, Guilin University of Electronic Technology, Guilin 541004, China; zhaoj6130@163.com (J.Z.); ayangbai@163.com (B.Y.)
2   Construction Institute, Guangdong Technology College, Zhaoqing 526000, China
\*   Correspondence: wangfeicheng@gdlgxy.edu.cn (F.W.); guidianma@163.com (B.M.)
†   These authors contributed equally to this work.

**Abstract:** The design of intersecting nodes in high-rise oblique mesh structures is a critical issue. The existing research on the intersecting nodes of oblique meshes mainly focuses on plane intersecting nodes and monotonic axial compression loads. The plane intersecting nodes cannot consider the contribution of the node's out-of-plane angle and floor beam to the node's out-of-plane stiffness in actual structures. In this paper, numerical analysis using ABAQUS was conducted to investigate the mechanical performance of space intersecting nodes of oblique meshes (OMSIN) under cyclic axial tension and compression loads, to provide a reference for the engineering application of oblique mesh structures in seismic regions. Six parameters were considered: the space intersecting angle, the plane angle symmetry coefficient, the plane intersecting angle, the out-of-plane constraint restraint, the steel content of the cross-section, and the concrete strength. The study showed that changes in the thickness of the steel tube wall are unfavourable for the uniform transmission of stress. Increasing the space intersecting angle significantly weakened the seismic performance, and the space angle affects the failure mode of the node. Asymmetric arrangements of the upper and lower plane angles caused nonlinear development of out-of-plane. The ultimate load and overall compressive stiffness of the specimen were positively correlated with the plane angle, and vertical constraints should be applied to the node position of components with plane angles greater than or equal to 70°. The out-of-plane constraint was a key factor affecting the seismic performance of the node, and it was proportional to the ultimate load of the component. In structural design, if the aim is to improve the mechanical performance of the component by increasing the steel content, more enormous out-of-plane constraints should be set to control plane external displacement strictly. The concrete strength is proportional to the ultimate axial load and axial stiffness, and its influence on the mechanical performance in the axial tension direction is not significant. Finally, a dimensionless skeleton curve model of the node was established. The existing formula for the bearing capacity of CFST columns was fitted to obtain the calculation formula for the axial yield and ultimate load of the OMSIN under cyclic loads.

**Keywords:** oblique mesh; intersecting nodes; CFST; numerical analysis; seismic performance

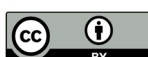

## 1. Introduction

With the acceleration of urbanization, modern high-rise buildings are rapidly developing with a new appearance [1–3]. As the building height increases, the lateral stiffness of the structure becomes a vital issue in the design. Under the conditions of ensuring structural safety and economy, selecting an economical, efficient, and effective structural system that can control the lateral displacement of high-rise structures became a new trend. The oblique mesh structure is a form of a structure composed of intersecting steel pipes and nodes. It has the advantages of a simple structure, high stiffness, and good

seismic performance and was widely used in high-rise buildings [4–9]. The intersecting node is formed by the intersection of four diagonal steel pipes in space. The existence of intersecting nodes will cause stress concentration at the node, thereby affecting the stability and safety of the structure. Therefore, the design of intersecting nodes in oblique mesh structures is essential. Therefore, studying the problem of intersecting nodes has important theoretical and practical significance in civil engineering.

In recent years, scholars from various countries made specific achievements in the research of oblique mesh structures. Zhao et al. [10] conducted research on the mechanical properties of intersecting nodes of diagrid structures composed of steel fibre-reinforced recycled concrete columns (OGSFRCIN). The study showed that OGSFRCIN has good mechanical properties and can be applied to engineering practice. Gao et al. [11] conducted experimental and numerical studies on the post-fire performance of steel tube T-joints filled with concrete. The results showed that due to the support provided by the core concrete to the steel tube, the failure mode of the node was plasticization of the steel tube rather than local buckling of the steel tube wall. Kong et al. [12] conducted finite element analysis on the KK-intersecting nodes and analyzed the difference between the nodes with and without stiffening ribs, providing a reference for practical engineering. Fan et al. [13] used ABAQUS to analyze the influence of different parameters on the mechanical behaviour of steel tubes intersecting nodes, considering cumulative damage. Zhou et al. [14] proposed a theoretical formula for the bearing capacity of spatially connected diagrid systems. Anupama et al. [15] conducted finite element analysis on the nodes formed by the connection of steel tube concrete columns and RC beams, mainly studying the axial compression performance of the nodes. Li et al. [16] found that the intersecting connection can fully develop plastic deformation during the loading process and analyzed the influence of different parameters on its mechanical properties, finally giving design suggestions. Shi et al. [17] used the stiffness equivalence principle to solve the critical problems in the preliminary design of high-rise diagrid structures: optimization of the intersecting angle and evaluation of the shear lag effect. Jun et al. [18,19] summarized the influence of anti-lateral stiffness-related factors on the plastic development process, stress distribution, and stiffness degradation of diagrid structures. Huang and his team [20–25] proposed a new type of node for diagrid structures and conducted research on it using experimental, numerical, and theoretical analysis methods.

In summary, existing research on oblique mesh intersecting nodes (OMIN) mainly focused on the mechanical properties of plane intersecting nodes under monotonic loads. At the same time, there is less research on the mechanical properties of nodes under reciprocating loads [26–28]. Moreover, plane intersecting nodes cannot consider the contribution of the node's out-of-plane angle and floor beam to the node's out-of-plane stiffness in actual structures. In order to fill the research gap mentioned above, this paper conducts research on the seismic performance of oblique mesh space intersecting nodes (OMSIN), with control parameters mainly including six parameters: space intersecting angle, plane angle symmetry coefficient, plane intersecting angle, out-of-plane constraint, steel content of the cross-section, and concrete strength. ABAQUS.2022 finite element software was used to establish the three-dimensional solid model of the node, and the reciprocating axial tension and compression loading was applied to the node to study the influence of different parameter settings on mechanical properties such as hysteresis curve, skeleton curve, stiffness degradation, energy dissipation capacity, out-of-plane displacement, Mises stress and strain, failure location and mode. The degraded bi-linear and tri-linear models were selected for axial compression and tension directions, respectively, and a dimensionless skeleton curve model was established. Finally, the existing formula for the ultimate bearing capacity of CFST columns was fitted and modified to obtain the calculation formula for the axial yield and ultimate load of OMSIN under cyclic loading. The aim was to provide a reference for the design and engineering application of intersecting nodes of diagrid structures in high-rise buildings.

## 2. Node Construction

The CFST-space intersecting node structure in oblique mesh proposed by Huang [21] was adopted. The structure was formed by four CFST columns intersecting at a certain angle in space. The four steel-tubed concrete columns of the node were cut along the space intersecting line and welded to an elliptical connecting plate. Concrete was poured inside the steel tubes. The main connecting and restraining components were the elliptical connecting plate, circumferential reinforcing ring, and circumferentially arranged stiffening ribs for out-of-plane restraint. All steel components were welded, as shown in Figure 1, where O in Figure 1a is the point for extracting the node's out-of-plane and vertical displacement.

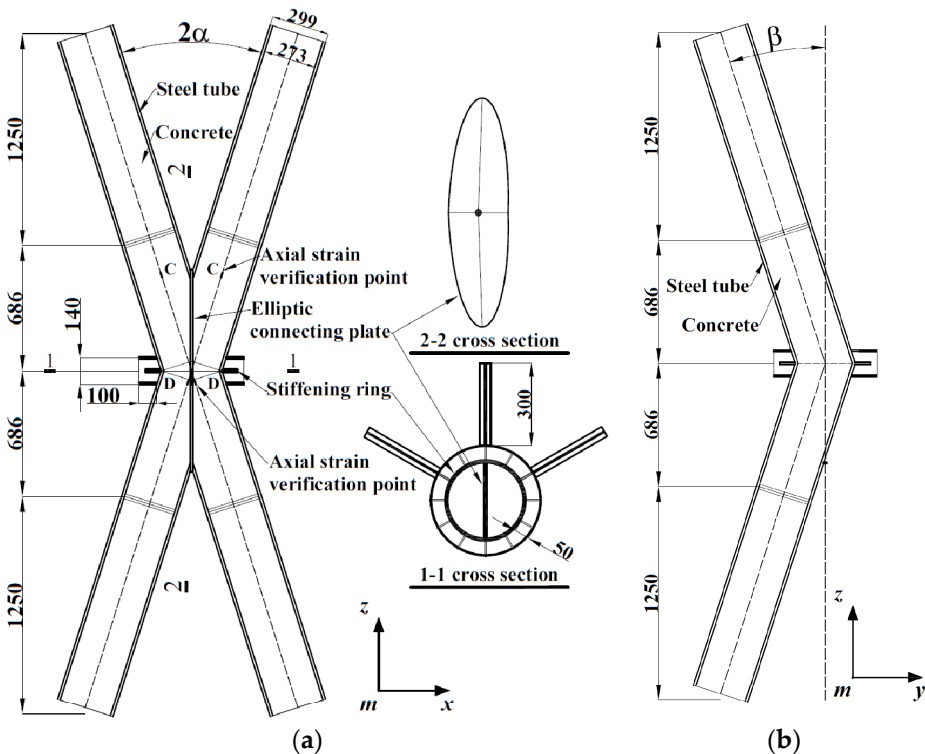

**Figure 1.** Oblique Mesh Intersecting Nodes Structure ($2\alpha$ is plane oblique angle; $\beta$ is space oblique angle). (**a**) Front view. (**b**) Lateral view.

## 3. Finite Element Model

### 3.1. Model Building and Boundary Condition Determination

The existing finite element models for the OMIN have certain limitations. The steel components, such as steel tubes, reinforcing rings, and elliptical connecting plates, are all modelled using shell elements. In Abaqus, shell elements are two-dimensional finite elements commonly used to analyze thin shell structures, such as metal sheets and plastic films. As the name suggests, shell elements are shell-shaped with a thin thickness defined by two nodes. The applicability of shell elements depends on their geometric shape. Generally, shell elements are suitable for the following situations: (1) structures with minimal thickness. Very thin structures such as wall panels, roofs, and panels; (2) a structure with a specific curvature. Shell elements are suitable for structures with large curvature; (3) the shell element is one of the best choices for planar stress state structures; The direction of action of the load acting on the structure is usually in one of the two principal planes. It should be noted that the geometric shape and thickness of shell elements should match the actual situation of the structure; otherwise, the calculation results will be inaccurate. In summary, the applicable conditions for shell elements are that the one-dimensional size (thickness direction) is much smaller than the other two-dimensional sizes (height and

width directions), and the deformation perpendicular to the thickness direction can be ignored. However, the steel components such as reinforcing rings, elliptical connecting plates, and stiffening ribs do not meet the above conditions, and the deformation in the thickness direction is also a research focus. Therefore, this paper establishes a three-dimensional solid model (C3D8R) using ABAQUS.2022 finite element analysis software. To ensure the accuracy of the model calculation results, the model was divided into structured meshes using meshing techniques, as shown in Figure 2. Considering that the specimens are prone to bulging and failure in the node area during the experiment, the mesh in the node area was refined.

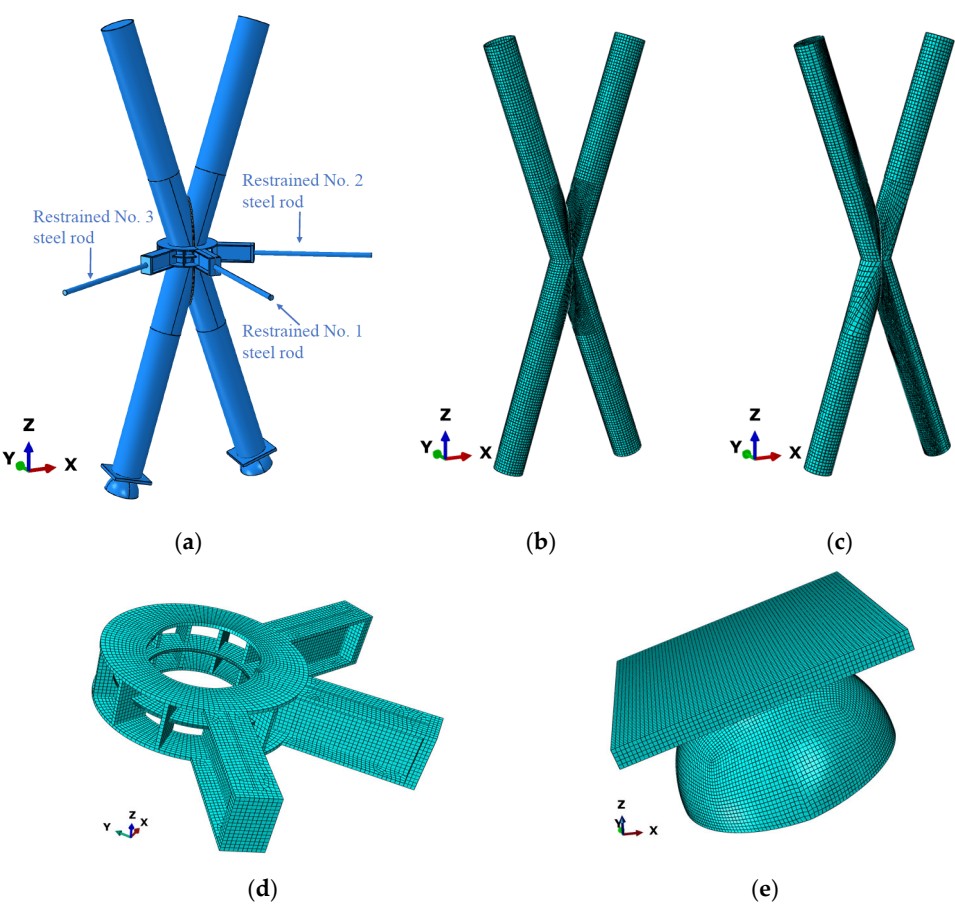

(a)      (b)      (c)

(d)      (e)

**Figure 2.** Finite Element Model. (**a**) Overall model. (**b**) Mesh division of external steel pipes. (**c**) Mesh division of core concrete. (**d**) Circumferential constraints. (**e**) Backing plate and spherical hinge.

Due to the tangential bond-slip between steel tubes and concrete in practical situations, cohesive elements are used to simulate the bond-slip effect in the tangential direction, while hard contact is used in the normal direction to prevent element penetration. In the actual situation, two steel columns on the same side of the connecting plate are connected by groove welding, and the connection between the node and the upper and lower limb steel tube concrete columns is achieved by inner lining pipe welding. Therefore, the contact between all steel components is set as a tie constraint in the model. Axial displacement is applied along the axis of the steel tube at the top of the steel tube to extract the load data for the steel tube and the core concrete by establishing a common reference point. To ensure that the steel tube is subjected to zero bending moment at the bottom during the experiment, a ball joint is used to connect the bottom, as shown in Figure 2e. When the component undergoes local bulging or significant lateral deformation, it can be considered a failure.

*3.2. Material Constitutive*

The constitutive model of core concrete proposed by Han [29,30] introduces the constraint effect of external circular steel pipes on the core concrete. This model considers the improvement effect of circumferential constraints on the compressive strength of the core concrete and corrects the softening behaviour of the descending section. It is often used in the numerical simulation of circular steel pipe concrete. Therefore, this article adopts its proposed stress–strain relationship for steel pipe-constrained core concrete., as shown in Formulas (1)–(7):

$$y = \begin{cases} 2x - x^2 \ (x \le 1) \\ \dfrac{x}{\kappa(x-1)^2 + x} \ (x > 1) \end{cases} \tag{1}$$

where:

$$x = \varepsilon/\varepsilon_0 \ ; y = \sigma/\sigma_0 \tag{2}$$

$$\kappa = \left(2.36 \times 10^{-5}\right)^{\left(0.25 + (\theta - 0.5)^7\right)} \times f_c \times 0.5 \ge 0.12 \tag{3}$$

$$\sigma_0 = \left[1 + \left(-0.054\theta^2 + 0.4\theta\right)\left(\frac{24}{f_c}\right)^{0.45}\right] f_c; f_c = 0.8 f_{cu} \tag{4}$$

$$\theta = A_s f_y / A_c f_c \tag{5}$$

$$\varepsilon_0 = \varepsilon_{cc} + 800 \cdot \theta^{0.2} \cdot 10^{-6} \tag{6}$$

$$\varepsilon_{cc} = \left(1300 + 12.5 f_c\right) \cdot 10^{-6} \tag{7}$$

where $\sigma$ and $\varepsilon$ represent the stress and strain values of concrete, respectively, and $f_{cu}$ represents the compressive strength of concrete cubes, $f_c$ represents the compressive strength of cylindrical specimens with different percentages of recycled aggregate replacement, $f_y$ represents the yield strength of steel, $A_s$ and $A_c$ represent the cross-sectional areas of the steel tube and concrete, respectively, and $\theta$ represents the confinement effect coefficient.

The constitutive relationship of the outer steel pipe and the steel components used for constraint in the node area adopts a five-segment quadratic plastic flow model [31]. The material properties of the steel are shown in Table 1, and the base plate and spherical hinge are set as rigid bodies.

**Table 1.** Mechanical Properties of Steel.

| Steel Type | Model Number | Elastic Modulus $E_s$/MPa | Yield Strength $f_y$/MPa | Ultimate Strength $f_u$/MPa |
|---|---|---|---|---|
| Outer steel pipe | Q345B | $2.06 \times 10^5$ | 380 | 574 |
| Elliptic connecting plate | Q345B | $2.06 \times 10^5$ | 365 | 570 |
| $\phi 42$ steel bar | 40Cr | $2.06 \times 10^5$ | 785 | 930 |

*3.3. Reliability Verification of Finite Element Model*

The plane and space intersection angles of the components used for verification are 35° and 1°, respectively, and the thickness of the connecting plate is 17 mm. The strength grade of the node concrete is C90, and that of the non-node area concrete is C70. Figure 3 shows the comparison between the load–displacement curves of the experiment and the finite element model. As shown in Figure 3a, the evolution of the load–displacement curves of the two is basically consistent before the peak load. Both curves show linear growth before the displacement reaches 8 mm, indicating that the components are in the elastic stage. After the displacement exceeds 8 mm, the slope of the curve decreases slightly until the peak. However, the experimental curve stops loading immediately after reaching the peak load, so there is no downward section consistent with the finite element curve. The elastic stiffness and peak load of the finite element component are slightly higher than those of the experimental component. This is because, in actual situations, the concrete contains a certain amount of porosity, which causes differences between the concrete in the actual situation and the homogeneous solid in the finite element model. In addition, the peak load corresponding to the experimental curve is slightly larger than that of the finite element model since the concrete in the actual situation is a porous and non-uniform material and a more significant displacement is required to close the initial gap. According to Figure 3b, the finite element curve shows linear growth in out-of-plane displacement, while the experimental curve shows a sudden change in slope at an axial displacement of 7 mm. This is because the axial compression loading of the steel tube in the experiment cannot be guaranteed to be absolutely along the axis of the steel tube; i.e., there is a certain eccentricity, which causes the slow development of lateral displacement in the early stage and a sudden increase in lateral displacement in the later stage. However, overall, the maximum difference between the out-of-plane displacement of the experiment and the finite element is only 1.27%, which is in good agreement.

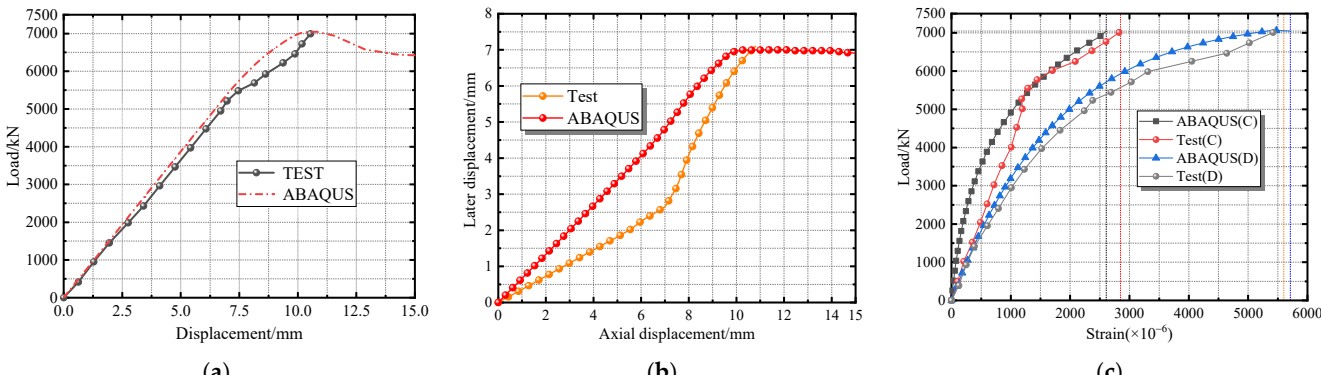

**Figure 3.** Comparison between Test Curve and Finite Element Curve. (**a**) Axial displacement-load curve. (**b**) Axial displacement-lateral displacement curve. (**c**) Load-axial strain curve.

As shown in Figure 3c, the strain-load curves of the two measuring points have similar evolution patterns. Due to the ideal nature of the model in the finite element analysis, the slope of the finite element curve is more significant than that of the experimental curve. However, the peak strains at points C and D differ by 8.49% and 2.00%, respectively. Overall, the finite element model and the experimental results show good agreement.

Figure 4 shows a comparison between the Mises stress cloud map of the finite element model and the experimental situation. As shown in Figure 4a–d, both the experimental and finite element components experience two types of failure: buckling failure of the steel tube in the node area and bending failure of the steel tube in the non-node area. According to Figure 4c, when the wall thickness and concrete strength of the steel tube in the node area and non-node area are inconsistent, the Mises stress shows non-uniformity along the boundary line, indicating that changes in wall thickness are not conducive to

the uniform transmission of stress. This may be one of the essential reasons for the large bending deformation of the steel tube in the non-node area. The buckling phenomenon of the steel tube in the node area is more obvious, while the steel tube reinforced by the circumferential reinforcement does not show obvious deformation. As shown in Figure 4e,f, the Mises stress of the circumferential reinforcement, stiffeners, and elliptical connection plate is far less than the tensile strength of the steel, and the stress distribution is uniform and symmetrical, indicating that they can still provide good constraints and reflect the restrictive effect of circumferential constraints on the component in the experiment. In summary, the finite element model can well reflect the interaction between the components in the experiment, and this modelling method can be used for further analysis and research on intersecting nodes.

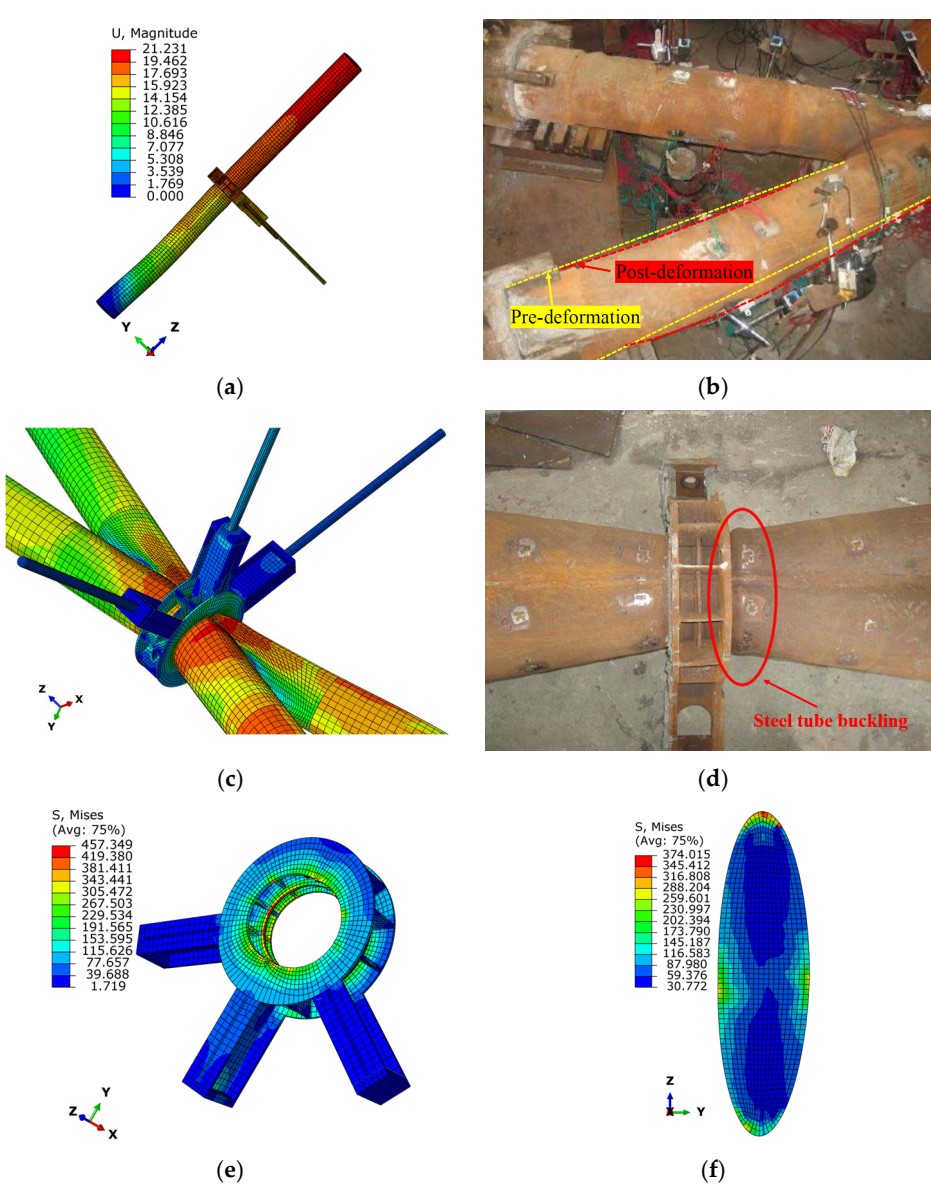

**Figure 4.** Comparison of Failure Modes between Finite Element Model and Test Component. (**a**) Failure pattern of finite element model. (**b**) Bending failure of lower extremity steel pipe of test member. (**c**) Steel pipe bulging in joint area of finite element model. (**d**) Steel pipe bulging in the joint area of test member. (**e**) Mises stress of restrained steel parts. (**f**) Mises stress of elliptic connecting plate.

### 4. Influence of Different Parameters on Mechanical Properties of OMSIN

*Parameter Setting*

This section mainly considers the influence of six parameters, namely, the space intersecting angle, symmetry coefficient, plane intersecting angle, out-of-plane constraint (ratio of the diameter of the out-of-plane restraint steel bar to the outer diameter of the steel tube), steel content, and concrete strength, on the mechanical properties of space intersecting nodes. The symmetry coefficient refers to the ratio of the upper limb angle to the lower limb angle in the plane, mainly examining the effect of the unequal intersection angle of the upper and lower limb steel tube concrete planes in actual engineering applications due to architectural aesthetics. The steel content is achieved by changing the wall thickness of the steel tube. A control group specimen is specified, and the other specimens change one parameter based on this specimen to explore the influence of variables on mechanical properties. The specific parameters are shown in Table 2. To save calculation time, the following optimizations are made to the test components: (1) The wall thickness of the column in the node and non-node areas of the test is set to the same size, and the concrete strength grade is consistent; (2) The spherical hinge base is removed, and the restraint method at the end of the lower limb steel tube concrete is fully fixed; (3) The out-of-plane restraint steel bar only restrains displacement along the axis of the steel bar. The load–displacement hysteresis curve under cyclic loading can well reflect the evolution process of the component from loading to failure. Since the study focuses on the mechanical properties of the node, the displacement of the hysteresis curve is taken as the vertical displacement of the O point at the node's centre. The vertical displacement-lateral displacement hysteresis curve can well reflect the development of the out-of-plane displacement during loading, and the displacement is taken as the vertical displacement and plane displacement of the O point at the centre of the node, as shown in Figure 1a. This can be used as a basis for considering whether the component undergoes significant deformation failure and whether the upper and lower limb steel tube concrete undergoes asymmetric deformation.

**Table 2.** Parameter Settings of Test Pieces.

| Parameter Name | Space Angle ($\beta$) | Plane Angle ($2\alpha$) | Symmetry Coefficient ($\Psi$) | Steel Content ($\varphi$) | Out-of-Plane Constraint ($\omega$) | Concrete Strength ($f_c$) |
|---|---|---|---|---|---|---|
| Standard group | 3° | 35° | 1.00 (35°/35°) | 0.13 | 0.15 | 90 MPa |
| Change group | 1° | 20° | 0.70 (35°/50°) | 0.05 | 0.00 | 30 MPa |
| | 2° | 50° | 0.50 (35°/70°) | 0.20 | 0.05 | 50 MPa |
| | 4° | 70° | 0.39 (35°/90°) | 0.25 | 0.10 | 70 MPa |
| | 5° | 90° | - | - | 0.20 | 120 MPa |

### 5. Results

Table 3 presents the finite element analysis results, in which "Node-Standard" was the standard group. "Node-Space1" was the specimen whose space angle was 1°. "Node-Sym0.70" was the specimen whose symmetry coefficient was 0.70; "Node-Plane20" was a specimen with a plane angle of 20°; "Node-Con0.00" was the specimen whose out-of-plane constraint coefficient was 0; "Node-Steel0.05" was the specimen with a steel content of 0.05; "Node-Concrete30" was a concrete specimen with concrete strength of 30 MPa.

**Table 3.** Main results of finite element model.

| Control Parameters | Component Name | $F_y$ (+) /kN | $F_y$ (−) /kN | $F_u$ (+) /kN | $F_u$ (−) /kN | $\delta_u$ /mm | $\Theta$ /(kN·m) | $k_c$ |
|---|---|---|---|---|---|---|---|---|
| Standard group | Node-Standard | 5890.10 | 4222.78 | 6222.11 | 5200.26 | 1.91 | 397.22 | 0.830 |
| Space angle | Node-Space1 | 5886.34 | 4222.65 | 6349.58 | 5141.26 | 0.83 | 373.67 | 0.847 |
| | Node-Space2 | 6029.15 | 4168.06 | 6503.35 | 5061.76 | 1.21 | 366.03 | 0.867 |
| | Node-Space4 | 5750.80 | 4004.15 | 6150.36 | 4698.79 | 36.18 | 373.65 | 0.820 |
| | Node-Space5 | 5520.37 | 3033.03 | 5531.21 | 4117.00 | 54.25 | 327.09 | 0.738 |
| Symmetry coefficient | Node-Sym0.70 | 5977.81 | 4044.65 | 6570.09 | 4871.63 | 33.92 | 415.31 | 0.876 |
| | Node-Sym0.50 | 5304.08 | 4013.65 | 6242.73 | 4210.26 | 33.39 | 490.71 | 0.833 |
| | Node-Sym0.39 | 4406.12 | 3213.28 | 5478.11 | 3913.25 | 6.40 | 455.27 | 0.731 |
| Plane angle | Node-Plane20 | 5628.60 | 4130.05 | 6199.52 | 5246.72 | 1.95 | 456.59 | 0.827 |
| | Node-Plane50 | 6275.72 | 4156.00 | 6697.62 | 4977.11 | 1.81 | 336.06 | 0.893 |
| | Node-Plane70 | 6513.06 | 4372.05 | 7186.35 | 5451.78 | 1.56 | 535.89 | 0.959 |
| | Node-Plane90 | 6234.46 | 4316.43 | 7111.80 | 5289.62 | 1.09 | 542.79 | 0.949 |
| Out-of-plane constraint | Node-Con0.00 | 4529.30 | 3847.95 | 4709.06 | 4651.71 | 102.90 | 464.23 | 0.628 |
| | Node-Con0.05 | 5350.91 | 3775.79 | 5401.88 | 4606.21 | 50.07 | 330.19 | 0.721 |
| | Node-Con0.10 | 5888.68 | 3873.45 | 6333.64 | 4681.38 | 46.88 | 352.01 | 0.845 |
| | Node-Con0.20 | 6024.17 | 4275.48 | 6314.38 | 5179.72 | 1.32 | 395.24 | 0.842 |
| Steel content | Node-Steel0.05 | 4578.04 | 1499.75 | 4910.12 | 1843.28 | 1.20 | 367.54 | 0.823 |
| | Node-Steel0.20 | 7552.80 | 6272.14 | 9026.46 | 7853.92 | 45.20 | 591.60 | 0.976 |
| | Node-Steel0.25 | 8463.85 | 7520.42 | 10,377.04 | 9079.87 | 39.90 | 725.48 | 0.893 |
| Concrete strength | Node-Concrete30 | 4641.21 | 4113.87 | 5336.96 | 5100.29 | 1.44 | 748.13 | 1.092 |
| | Node-Concrete50 | 5520.37 | 4108.39 | 5576.77 | 5147.45 | 1.55 | 728.89 | 0.971 |
| | Node-Concrete70 | 5492.21 | 4140.10 | 5929.77 | 5156.74 | 1.63 | 752.17 | 0.947 |
| | Node-Concrete120 | 6463.56 | 4166.67 | 7004.76 | 5193.63 | 2.15 | 806.57 | 0.792 |

Note: $F_y$ denotes the yield load of the node, and the yield point is defined according to references [32,33]. The method for determining the yield point is as follows: the straight line passing through the origin and 0.75 $F_u$ point intersects with the horizontal line passing through the peak point of the curve at a certain point, which is the yield point, where (+) and (−) denote compression and tension along the axis, respectively. $F_u$ denotes the ultimate load of the node, and the interpretation in parentheses is the same as that of $F_y$. $\delta_u$ denotes the maximum displacement of the node in the plane, $\Theta$ denotes the total energy dissipation of the node, and $k_c$ denotes the strength enhancement coefficient [34]. $k_c = F_u/F_{u0}$, where $F_{u0} = A_s f_y + A_c f_c$ represents the simple superposition of the bearing capacity of the steel tube and concrete. It intuitively reflects the mutual compensation effect between the steel tube and the concrete.

## 6. Discussion

### 6.1. Space Intersecting Angle

Figure 5 shows the mechanical performance parameters of components with different space angles. As shown in Figure 5a, the hysteresis curve of all specimens was full, and there was no apparent pinching phenomenon, indicating that the contact and cooperation between the external steel tube and the core concrete of all specimens were good and that there was no apparent bond slip. When the space angle was ≤4°, the peak load of the specimens in the compression stage was not significantly different, and the descending segment after the peak load was not prominent, indicating a stable load-bearing capacity. Unlike the other specimens, the specimen with a space angle of 5° had a significant downward segment in the compression stage. Combined with Figure 5b,c, the influence of the space angle on the stiffness of the positive and negative elastic stages was minimal, and the skeleton curve was close to coincidence. However, the skeleton curve of the specimens with space angles of 4° and 5° fluctuated significantly in the tensile cyclic stage, and the stiffness degradation was significant. Especially for the specimen with a space angle of 5°, the stiffness in both positive and negative directions decreased significantly in the second

cycle. It unloaded significantly at an axial displacement of 7.5 mm in the third loading cycle. The skeleton curve dropped rapidly and reached the minimum load at an axial displacement of 12.5 mm, only 62.35% of the peak load. Then, the hysteresis curve stably bore the load and slightly rose. There were mainly two reasons for this: first, the specimens with space angles of 4° and 5° had larger out-of-plane displacements in Figure 5d, with maximum values of −36.1795 and 54.253, respectively, while the out-of-plane displacements of the other specimens developed linearly within the interval of (−5, 5), and the out-of-plane residual deformations were 9.28 and 13.36, respectively, which were significantly deformed. Secondly, the out-of-plane restraint bar in the 0° direction was broken, and then, the out-of-plane restraint bar in the ±45° direction began to play a role.

According to Figure 5e, the energy dissipation capacity of each component increased step by step and began to show differences after the second cycle. The energy dissipation capacity of the component with a space angle of 3° increased linearly, indicating that the component can still maintain a large energy dissipation capacity in the later large displacement cycles. However, the energy dissipation capacity of the component with a space angle of 5° increased more slowly after the fifth cycle, and the energy dissipation in the last cycle was only 28.39% of the total energy dissipation. The total energy dissipation was only 76% of the total energy dissipation of the component with a space angle of 3°. Components with a spatial angle of 4° still maintain a significant energy dissipation capacity throughout the entire loading stage, mainly due to their excellent load-bearing capacity and sizeable vertical displacement of nodes. However, it was prone to lateral instability and failure due to the difficulty in controlling the out-of-plane displacement. and it was not recommended to use components with space angles greater than or equal to 4° in engineering practice.

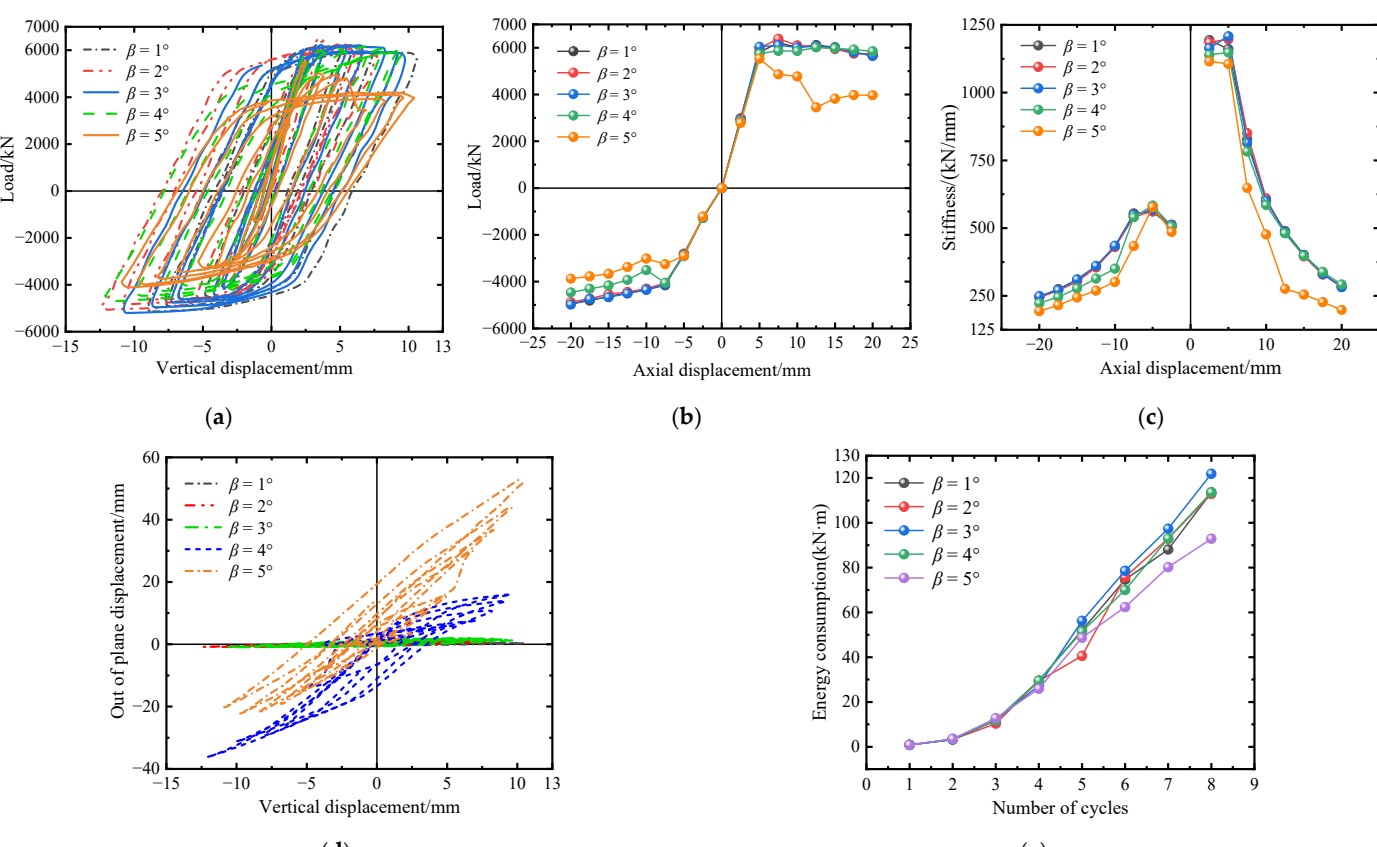

**Figure 5.** Relevant mechanical property parameter curves of components with different space angles. (**a**) Displacement load curve. (**b**) Skeleton curve. (**c**) Stiffness degradation curve. (**d**) Axial displacement-out of plane displacement curve. (**e**) Energy consumption curve.

Figure 6 shows the Mises stress and concrete compression damage of components at different space angles. As shown in Figure 6a–e, the maximum Mises stress values of all specimens reached the ultimate strength of the steel tube, for specimens with space angles ≤ 3°, the maximum stress value was mainly located in the middle of the elliptical connecting plate near the concave side and the non-nodal area far from the end. For specimens with space angles > 3°, the maximum stress value was mainly located on the concave side of the elliptical connecting plate, and the positive *y*-axis side of the loaded end steel tube, and the reason for the difference was mainly due to the different failure modes of the specimens. The failure mode of specimens with space angles ≤ 3°was mainly bending failure of the steel tube concrete column, while the failure mode of specimens with space angles >3°was mainly out-of-plane instability failure of the node. According to Figure 6f–h, the area of compression damage of the specimen with a space angle of 1° was relatively large, and the main damage area was the interval (105.6 mm, 1054.5 mm) from the starting point of the intersection. In the interval (−105.6 mm, 105.6 mm), the average compression damage was only 0.49, mainly due to the reinforcement ring setting in this area, which improved the radial constraint of the core concrete. For specimens with space angles of 4° and 5°, the compression damage was mainly concentrated on the concave side of the node area and the loaded end. The maximum compression damage value was observed in the specimen with a space angle of 5°, while the concrete damage in the non-nodal area was relatively small. This difference was mainly due to the maximum lateral instability degree of the component, and the bending moment on the steel tube concrete at the end was larger during failure.

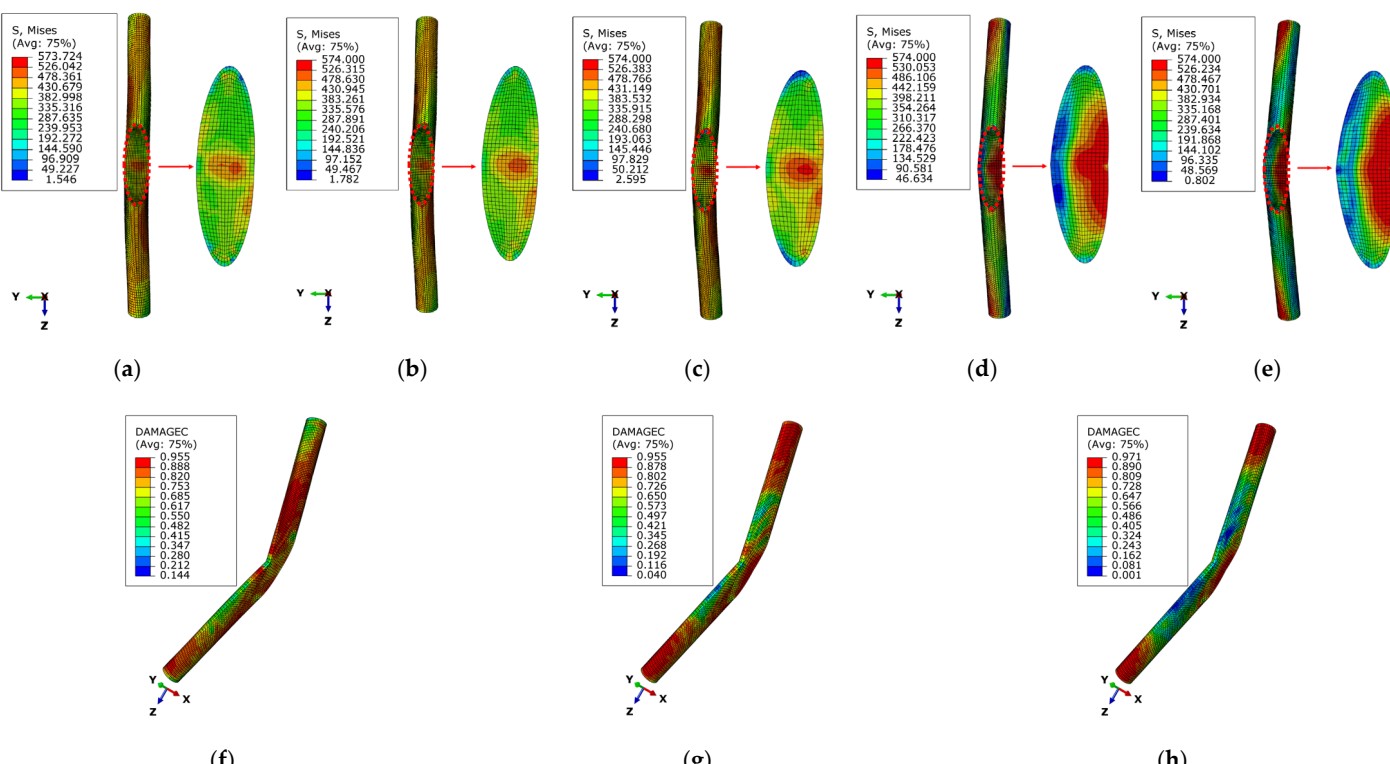

**Figure 6.** Mises stress and concrete compression damage of members with different space angles. (**a**) *β* = 1°. (**b**) *β* = 2°. (**c**) *β* = 3°. (**d**) *β* = 4°. (**e**) *β* = 5°. (**f**) *β* = 1°. (**g**) *β* = 4°. (**h**) *β* = 5°.

*6.2. Symmetry Coefficient of Plane Intersecting Angle*

The mechanical performance parameter curves of components with different plane symmetry coefficients are shown in Figure 7. As shown in Figure 7a–c, the ultimate bearing capacity of the node was negatively correlated with the plane angle symmetry coefficient. When the symmetry coefficient was ≥0.70, the degree of attenuation of the node's

ultimate bearing capacity was insignificant, and the descending segment of the hysteresis curve after the peak load was not noticeable, indicating an excellent load-carrying capacity. When the symmetry coefficient was <0.70, the degree of attenuation of the node's ultimate bearing capacity was significant. The descending segment of the hysteresis curve after the peak load was prominent, indicating an insufficient load-carrying capacity. It is worth noting that the vertical residual deformation of the node at the loading end of the component with a symmetry coefficient of 0.70 was as high as 12.8781 mm, and the hysteresis curve moved towards the positive direction of the *x*-axis as a whole. The component with a symmetry coefficient of 0.50 also exhibited the same trend. This indicates that the asymmetric arrangement of the plane angle will increase the vertical residual deformation of the node and weaken its safety. The tensile and compressive stiffness of the node was proportional to the symmetry coefficient. When the symmetry coefficient was ≥0.70, the degradation curves of the overall axial stiffness in the positive and negative directions tended to overlap, and the degradation rate was consistent. When the symmetry coefficient was less than 0.70, the positive overall axial stiffness decreased significantly, and the initial stiffness in the negative direction was larger than that of the other specimens. However, the rate of stiffness degradation increased in the later stages. This was mainly due to the asymmetry, which caused the lower limb steel tube concrete to be in a compression, bending, and shear stress state. The existence of bending moments and shear forces will weaken the component's axial tensile and compressive stiffness. In the later stage, due to the increase in axial displacement, the bending deformation of the lower limb steel tube concrete gradually increased with the increase in asymmetry. Therefore, the impact on the rate of stiffness degradation increased.

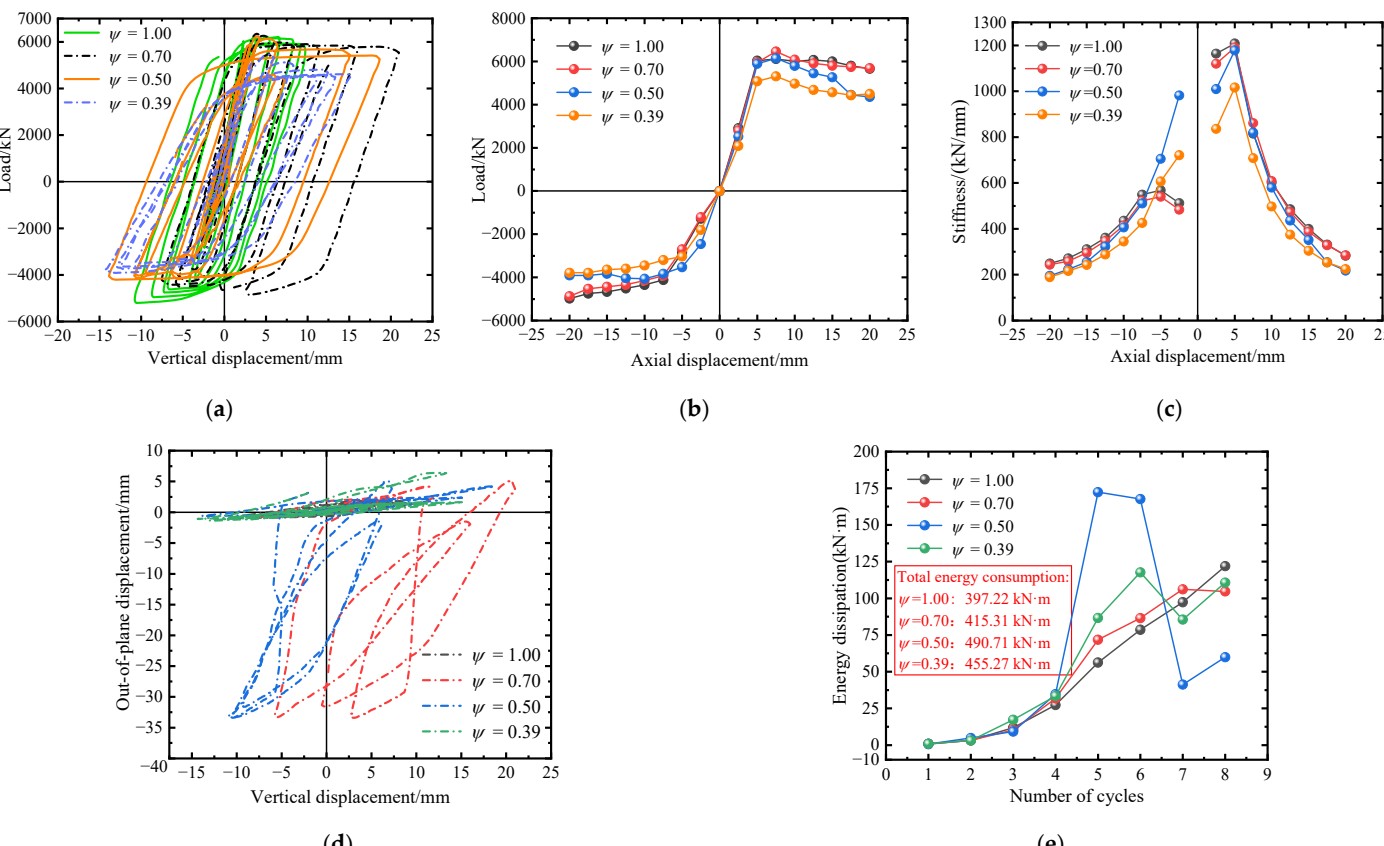

**Figure 7.** Relevant mechanical property parameter curves of members with different symmetry coefficients. (**a**) Displacement load curve. (**b**) Skeleton curve. (**c**) Stiffness degradation curve. (**d**) Axial displacement-out of plane displacement curve. (**e**) Energy consumption curve.

As shown in Figure 7d, the out-of-plane displacement of the nodes of the components with symmetry coefficients of 0.50 and 0.70 both exhibited nonlinear development in the later stages of loading, while the out-of-plane displacement of the nodes of the component with a symmetry coefficient of 0.39 was only 6.41 mm, which was still within a controllable range compared to the former two. This was mainly due to the significant difference in the plane angles of the upper and lower limbs of the component with a symmetry coefficient of 0.39, with the larger angle of the lower limb steel tube concrete experiencing a greater bending moment, resulting in bending failure at a lower load and smaller out-of-plane displacement. However, the out-of-plane displacement of components with a symmetry coefficient of 0.39 also showed a nonlinear growth trend, indicating that stricter out-of-plane constraints were required when the plane angles of the upper and lower limbs were asymmetric. Based on Figure 7a, it can be seen that in practical application, more enormous out-of-plane constraints and vertical deformation constraints should be set for nodes with inconsistent upper and lower limb plane angles. According to Figure 7e, when the symmetry coefficient was not equal to 1, the energy dissipation of the node increased in a zigzag pattern during the cyclic process, and the energy dissipation of the node with a symmetry coefficient of 0.50 experienced a sudden increase in the fifth cycle. Moreover, the total energy dissipation of all components with symmetry coefficients not equal to 1 was more excellent than that of the component with a symmetry coefficient of 1, mainly due to the more significant vertical deformation of the asymmetrical angle nodes.

Figure 8 shows the Mises stress cloud maps of components with different plane symmetry coefficients. As shown in the figure, the Mises stress distribution of all specimens' steel tube and elliptical connection plate was asymmetric in the vertical direction, with the extreme values distributed on the side with a larger plane angle. The failure mode of the components with symmetry coefficients of 0.50 and 0.39 was out-of-plane bending failure of the lower limb steel tube concrete (yoz plane), while the failure mode of the component with a symmetry coefficient of 0.70 was the in-plane bending failure of the lower limb steel tube concrete (xoz plane), and the steel tube bulging and tearing failure occurs at the lower limb steel tube node.

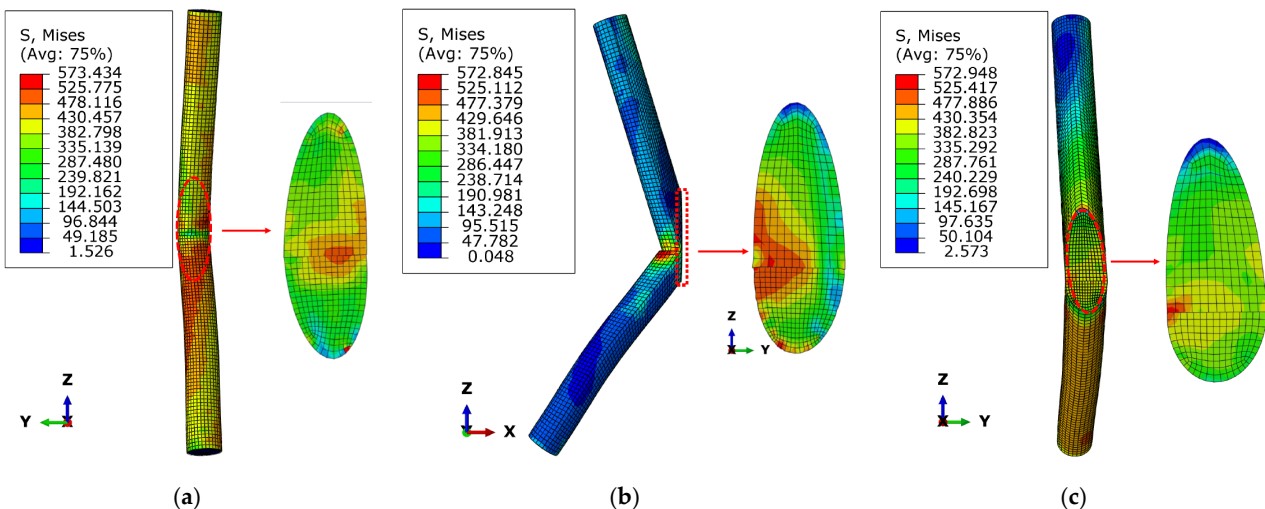

**Figure 8.** Mises stresses of members with different plane symmetry coefficients. (**a**) Symmetry coefficient = 0.70. **b**) Symmetry coefficient = 0.50. (**c**) Symmetry coefficient = 0.39.

According to the local stress cloud map, as the symmetry coefficient decreases, the degree of asymmetry of the Mises stress distribution of the elliptical connection plate about the origin increases. The maximum Mises stress distribution of the connection plate of the component with a symmetry coefficient of 0.39 is located in the lower middle part of the concave side, with an area of only about 2.88% of the total area of the connection

plate. The concave side of the connection plate of the component with a symmetry coefficient of 0.50 exhibits crushing. Therefore, it was observed that the failure occurred in the steel tube concrete on the side with a larger plane angle, and the primary failure mode was bending failure caused by a bending moment. The concave side of the elliptical connection plate of the component with a symmetry coefficient not equal to 1 should be thickened.

## 6.3. Plane Intersecting Angle

Figure 9 presents the mechanical performance parameters of components with different plane angles. As shown in Figure 9a,b, the specimens' ultimate load and overall compressive stiffness were positively correlated with the plane angle. The effect of the plane angle on the positive ultimate load was more significant than that on the negative ultimate load. The negative initial stiffness of all specimens had a significant difference. However, the difference in the stiffness degradation curve was slight in the later loading stage, indicating that the change in the plane angle had little effect on the tensile performance of the node. The positive residual loads of the specimens with plane angles of 70° and 90° were 0.854 and 0.868 of their ultimate loads, respectively. The degradation rate of positive stiffness was higher, and the load-bearing capacity was lower than other components. Notably, the hysteresis curves of these two types of components exhibited obvious left and right shifts, indicating that the plastic deformation and vertical residual displacement of the node were large. According to Figure 9d, except for the component with a plane angle of 35°, the energy consumption of the other components increased nonlinearly. Especially, the component with a plane angle of 50° showed a decrease in energy consumption capacity in the seventh cycle, and the total energy consumption was only 0.5269 of that of the component with a plane angle of 70°.

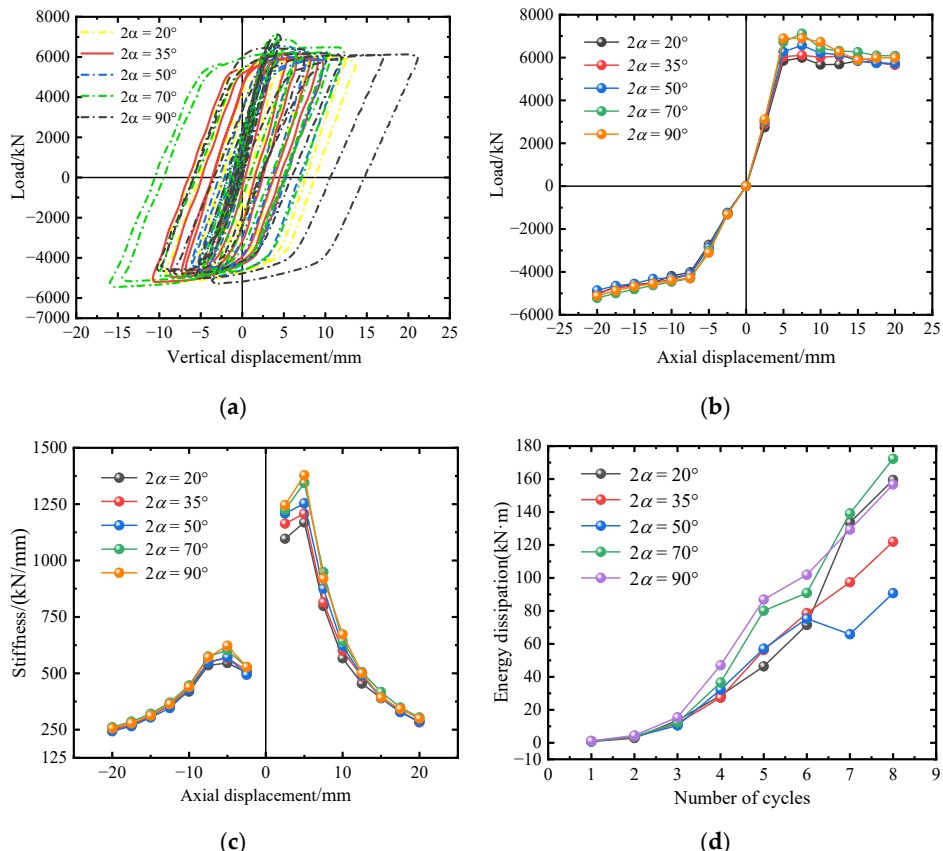

(a)

(b)

(c)

(d)

**Figure 9.** Relevant mechanical property parameter curves of members with different plane angles. (**a**) Displacement load curve. (**b**) Skeleton curve. (**c**) Axial displacement-out of plane displacement curve. (**d**) Energy consumption curve.

Figure 10 shows the Mises stress cloud map and overall displacement cloud map of specimens with different plane angles. It can be seen from the figure that there was no obvious bending of the member in the ultimate state of the component with a plane angle of 20°. The failure mode was the bulging of the steel tube in the node area. The components with plane angles of 70° and 90° experienced obvious bending failure, and the extreme displacement values mainly occurred in the CFST in the non-node area, which were 17.92 mm and 15.79 mm, respectively. The steel tube stress at this location reached its ultimate tensile strength, and plasticity develops fully, resulting in a large z-direction displacement at the node's centre, leading to large residual deformations of −7.19871 mm and 9.57204 mm, respectively. The failure of these two types of components occurred in the CFST in the non-node area, and the outer steel tube at the node was still in the elastic-plastic stage. The mean values of the Mises stress of the concrete were 31.39 MPa and 45.32 MPa, respectively, which still had a large safety margin. Therefore, the design criterion of "component failure before node failure" was satisfied. However, considering the safety of the node, vertical constraints should be applied to components with plane angles greater than or equal to 70°.

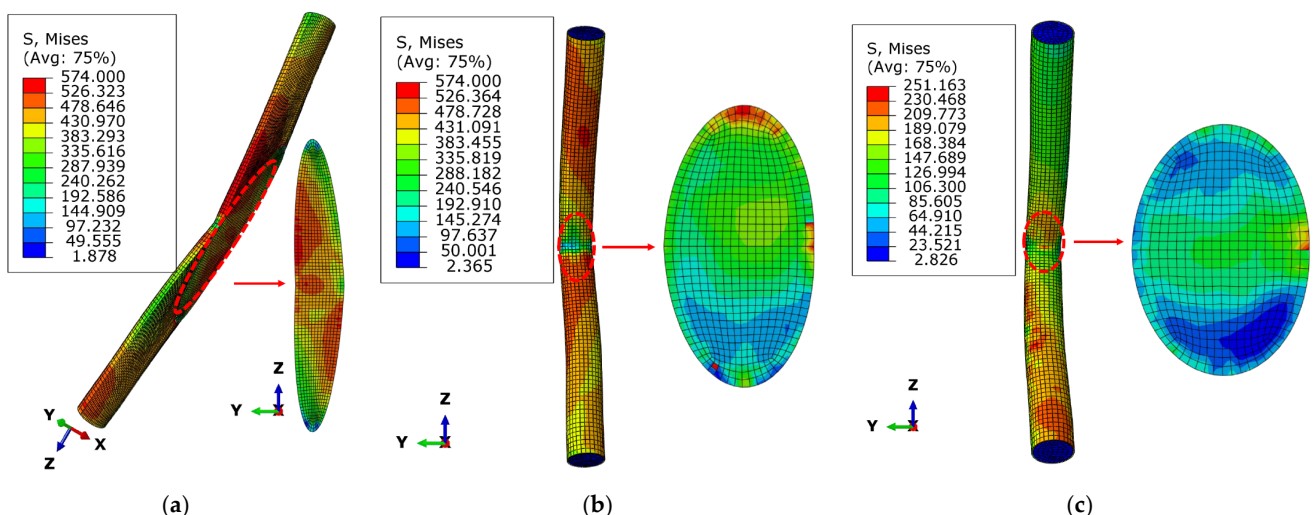

|  |  |  |
|---|---|---|
| (**a**) | (**b**) | (**c**) |

**Figure 10.** Mises stress nephogram and overall displacement nephogram of specimens with different plane angles. (**a**) $2\alpha = 20°$. (**b**) $2\alpha = 70°$. (**c**) $2\alpha = 90°$.

*6.4. Section Steel Content*

The curve diagrams of mechanical performance parameters for components with different steel contents are shown in Figure 11. As shown in Figure 11a, the steel content significantly influenced the hysteresis curve shape of the component. The hysteresis curve of the specimen with a steel content of 0.05 was shuttle-shaped, with a smaller area enclosed by the hysteresis loop, indicating a weaker energy dissipation capacity. Combined with Figure 11e, it can be seen that although the energy dissipation of the specimen with a steel content of 0.05 increased step by step, the increasing rate was relatively small compared to other components. This was mainly due to the smaller hoop coefficient, which cannot provide sufficient hoop restraint for the circumferential expansion of the steel tube on the core concrete, resulting in local bulging. The total energy dissipation of the specimens with steel contents of 0.20 and 0.25 was increased by 94.17% and 149.43%, respectively, compared to the specimen with a steel content of 0.05. This was partly due to the significant improvement of the tensile and compressive properties of the component with the increase in steel content and partly due to the nonlinear increase in the out-of-plane

displacement of the specimens with steel contents of 0.20 and 0.25 in the last cycle, increasing in the vertical displacement of the nodes.

As shown in Figure 11b, the steel content was positively correlated with the positive and negative ultimate loads and the overall stiffness of the component. The positive ultimate load of the specimen with a steel content of 0.25 was increased by 87.33% compared to that of the specimen with a steel content of 0.05, which was a significant improvement. However, it only increased by 3.21% compared to the specimen with a steel content of 0.20. This was partly due to the insufficient compressive strength of the core concrete and partly due to the insufficient out-of-plane restraint of the specimen with a steel content of 0.25, resulting in a sharp drop in the skeleton curve in the later stage of the loading cycle. As shown in Figure 11c, the ratio of residual stiffness to initial stiffness of the specimens with steel contents of 0.20 and 0.25 in the positive direction was 0.2712 and 0.2406, respectively, indicating that the larger the steel content, the greater the rate of stiffness decay in the later stage due to the sharp increase in the out-of-plane bending moment and displacement. Therefore, when aiming to improve the mechanical performance of the component by increasing the steel content in the structural design, larger out-of-plane restraints should be set to utilize the material properties fully.

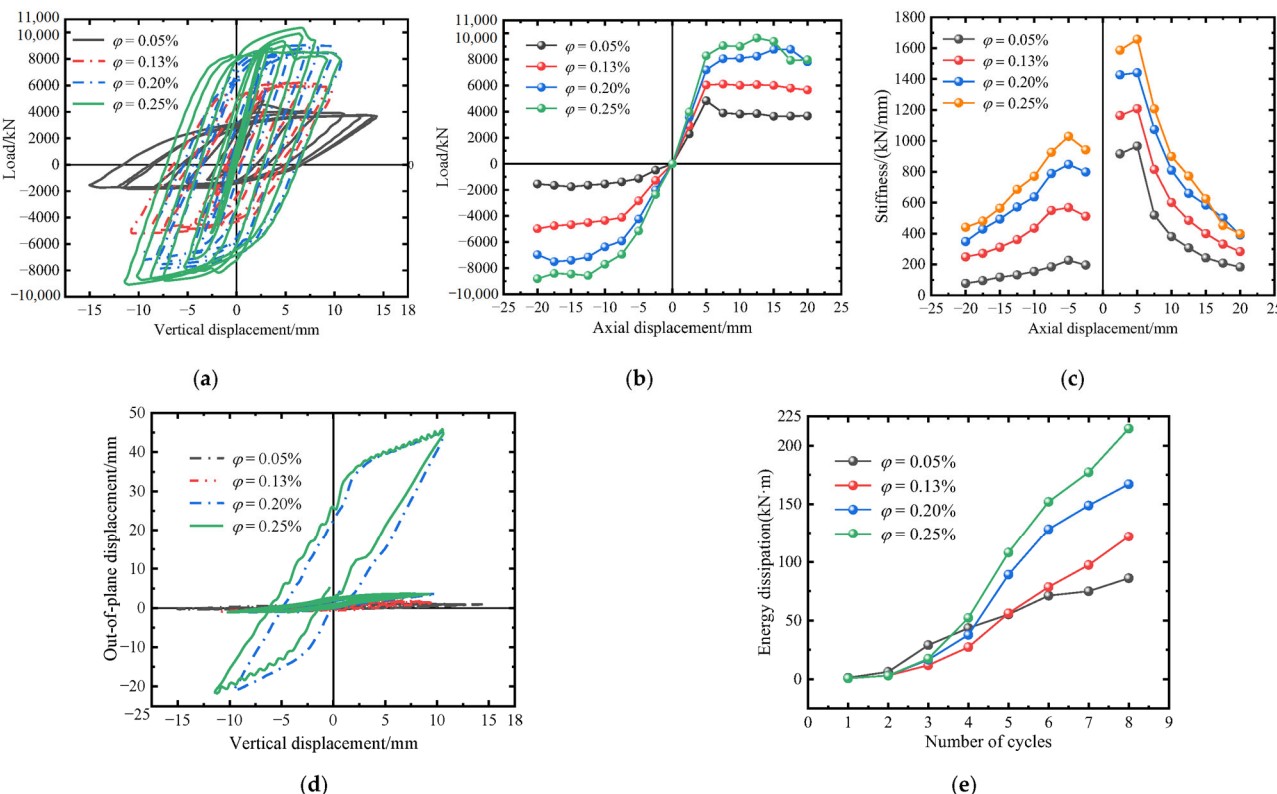

**Figure 11.** Relevant mechanical property parameter curves of members with different steel contents. (**a**) Displacement load curve. (**b**) Skeleton curve. (**d**) Axial displacement-out of plane displacement curve. (**e**) Energy consumption curve.

Figure 12 shows the Mises stress and compressive damage cloud maps of components with different steel contents. It can be observed that the specimen with a steel content of 0.05 exhibited a bulging and tearing failure at the end of the lower limb steel tube, while the specimens with steel contents of 0.20 and 0.25 failed due to instability caused by out-of-plane constraint failure. In the former case, the maximum stress occurred in the lower limb steel tube's middle and end regions, and the concrete's compressive damage reached its maximum value in these regions. This was mainly because, under axial compression loading, the lateral expansion of the core concrete at the end was restrained by

the hoop effect, resulting in a multi-directional high-stress state in which the steel tube is subjected to axial compression and hoop tension, ultimately leading to the crushing of the core concrete at the end and the tearing failure of the outer steel tube. In the latter two cases, the maximum stress occurs on the concave side of the elliptical connecting plate, mainly caused by the failure of out-of-plane constraint during the later loading stage.

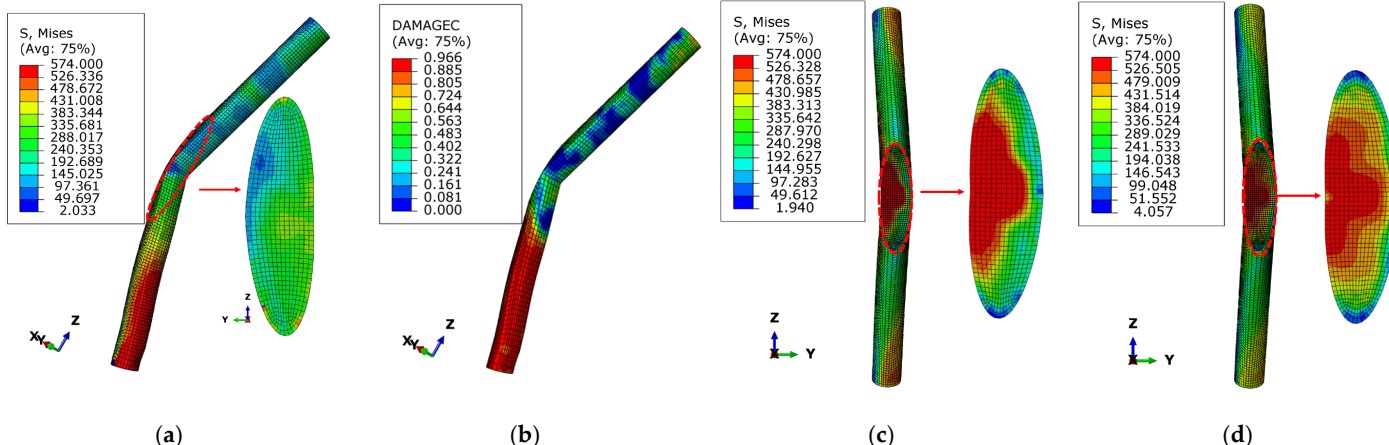

(a)  (b)  (c)  (d)

**Figure 12.** Mises stress nephogram and compression damage nephogram of members with different steel content. (**a**) φ = 0.05. (**b**) φ = 0.05. (**c**) φ = 0.20. (**d**) φ = 0.25.

*6.5. Out-of-Plane Constraints*

Figure 13 shows the parameter curves of specimens with different out-of-plane constraints. As shown in Figure 13a–c, the size of the out-of-plane constraint was proportional to the ultimate load of the component. The specimens with $\dot{\omega}$ values of 0.15 and 0.20 exhibited stronger load-carrying capacity during the axial compression cycle, and the descending segment of the curve after the peak was not apparent. The positive and negative skeleton curves tended to coincide, and the energy dissipation capacity was similar. This indicates that 0.15 was the limit point at which changes in out-of-plane constraints can improve the seismic performance of nodes. According to Figure 13c, the overall stiffness of the components with $\dot{\omega}$ of 0.15 and 0.20 was the largest, while that of the component with $\dot{\omega}$ of 0 was the smallest. The negative stiffness degradation curves of the components with $\dot{\omega}$ of 0 and 0.05 tended to coincide, indicating that changing $\dot{\omega}$ had no significant improvement on the node's tensile stiffness degradation rate within the range of $\dot{\omega} \leq 0.05$. When $\dot{\omega} \geq 0.15$, changing the size of the out-of-plane constraint had no significant improvement on the seismic performance of the component.

According to Figure 13d, the node with $\dot{\omega}$ of 0, i.e., without lateral constraint, exhibited nonlinear growth of vertical and out-of-plane displacement in the later stage of the cycle, with peak values of −16.15 mm and 102.90 mm, respectively. This was also the main reason for its sudden increase in energy dissipation capacity (Figure 13e). The specimens with $\dot{\omega}$ of 0.05 and 0.10 exhibited a significant descending segment in the positive direction of the hysteresis curve, and the latter exhibited a sudden drop-unloading phenomenon of the skeleton curve after the axial displacement was 10 mm, mainly due to the fracture of the lateral constraint bar in the 0° direction. The skeleton curve then stabilized and had an upward trend, mainly due to the lateral constraint in the ±45° direction beginning to constrain the out-of-plane deformation. Except for the unconstrained nodes that experienced large out-of-plane deformation failure, the energy dissipation capacity of the other specimens increased with the increase in $\dot{\omega}$, indicating that the out-of-plane constraint is a key factor affecting the seismic performance of the node.

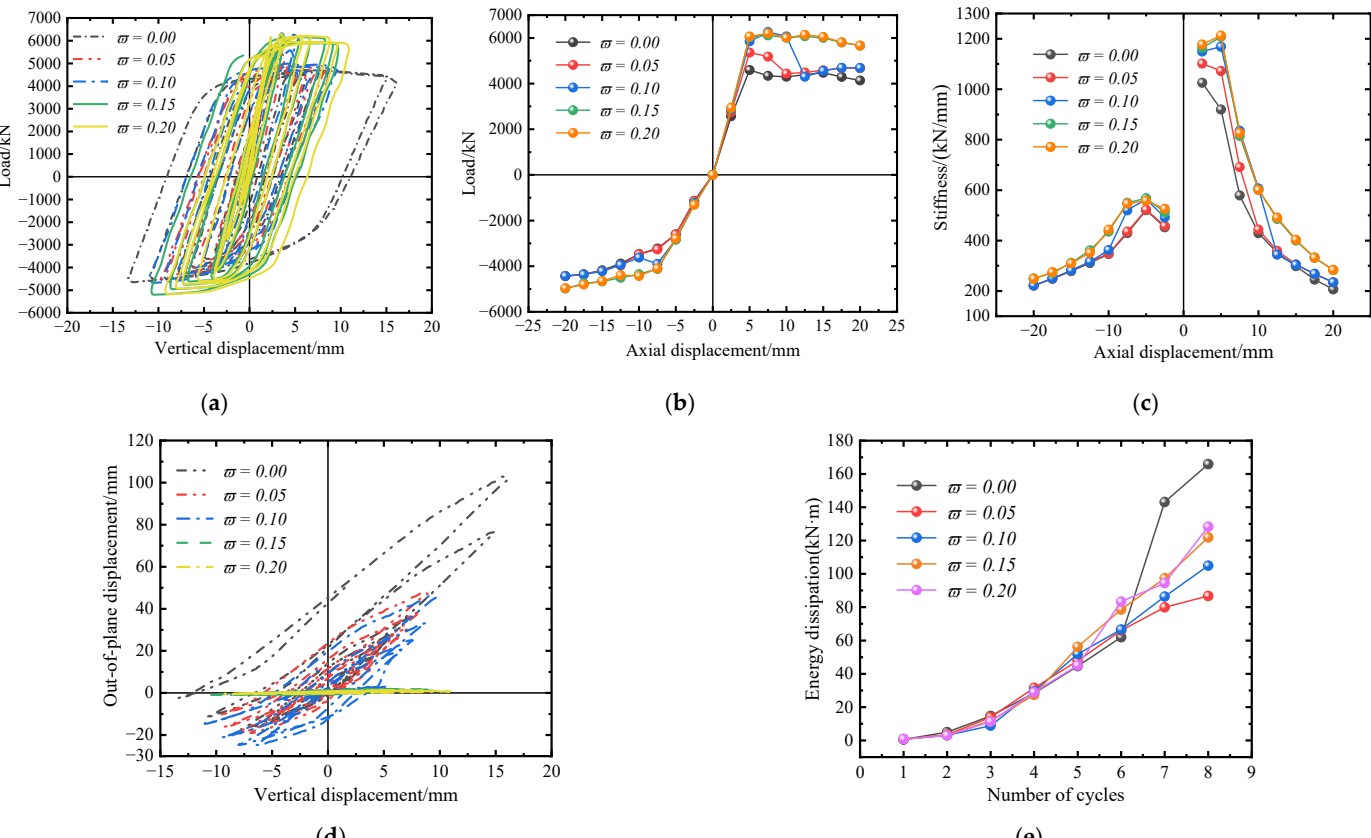

**Figure 13.** Relevant mechanical property parameter curves of members with different out of plane constraint sizes. (**a**) Displacement load curve. (**b**) Skeleton curve. (**c**) Stiffness degradation curve. (**d**) Axial displacement-out of plane displacement curve. (**e**) Energy consumption curve.

Figure 14 shows the Mises stress cloud maps of components with different out-of-plane constraint sizes. As shown in the figure, the two failure modes were large out-of-plane deformation failure and bending failure of the steel tube in the non-node area, respectively. The area of the maximum Mises stress of the component with $\acute{\omega}$ = 0 occurred on the concave side of the connecting plate, and the area of the extreme value was relatively large. The stress extreme value area of the component with $\acute{\omega}$ = 0.20 was in the steel tube of the non-node area, indicating that the out-of-plane constraint affects the failure mode and location of the component.

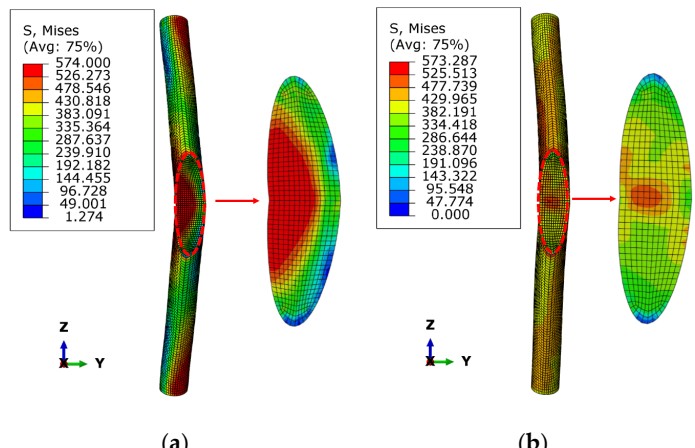

**Figure 14.** Mises stress nephogram of components with different out of plane constraint sizes. (**a**) $\acute{\omega}$ = 0.00. (**b**) $\acute{\omega}$ = 0.20.

### 6.6. Concrete Strength

Figure 15 shows the parameter curves of specimens with different concrete strength groups. As shown in Figure 15a, the concrete strength was proportional to the positive ultimate load. It had little effect on the negative ultimate load and the evolution law of the hysteresis curve. Combined with Figure 15b, it can be seen that the axial compression skeleton curve presented different evolution laws with the change in concrete strength. The peak point and the descending section of the curve of the specimen with concrete strength ≤ 50 MPa were not obvious, while the curve of the specimen with concrete strength > 50 MPa had a clear inflection point, peak, and descending section, which was mainly related to the brittleness of the specimen with higher concrete strength. All components' axial tension skeleton curves of all components had little difference, and the curves basically coincided. This was mainly because the tensile strength of concrete was weak. In the process of axial tension, almost only the outer steel tube bore the axial tension. The role of concrete is to delay the necking phenomenon of the external steel tube and improve the proportional limit and yield point of the steel tube, which is also the critical factor for the initial axial stiffness of components with higher concrete strengths to be slightly larger.

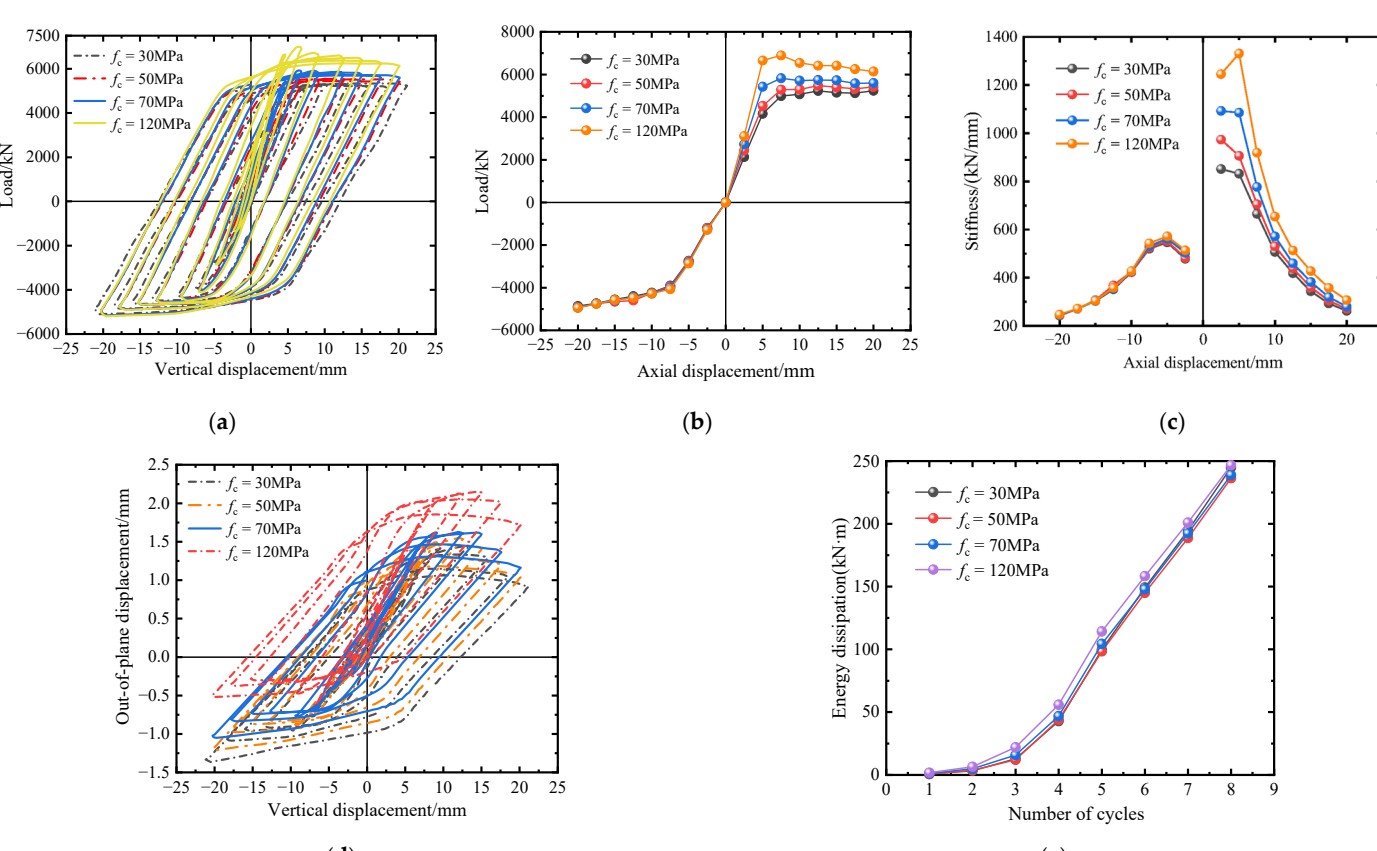

**Figure 15.** Relevant mechanical property parameter curves of members with different concrete strength. (**a**) Displacement load curve. (**b**) Skeleton curve. (**c**) Stiffness degradation curve. (**d**) Axial displacement-out of plane displacement curve. (**e**) Energy consumption curve.

According to Figure 15e, the energy consumption capacity of each specimen stage increased with the increase in concrete strength. The total energy consumption of the component with a concrete strength of 120 MPa was 31.70% higher than that of the component

with a concrete strength of 30 MPa, but the improvement was not very large. After analysis, it was found that this was mainly because when the concrete strength is higher, the core concrete needs a larger constraint, i.e., the hoop coefficient is larger. For stiffness degradation, the rate of stiffness decay in the later stage of the component with higher concrete strength increases due to its high brittleness, and the residual stiffness is not much different. For the out-of-plane displacement evolution, all specimens' vertical displacement-out-of-plane displacement curves have the same shape and evolution law without apparent fluctuation, indicating that the out-of-plane constraint condition is sufficient. Except for the curve of the component with a concrete strength of 120 MPa, all other curves evolve cyclically around the coordinate (0, 0). After analyzing the results of the finite element model, it was found that this was mainly because the non-node area steel tube concrete column of the component with a strength of 120 MPa undergoes slight bending in the early loading stage, resulting in out-of-plane residual deformation at the node. However, the maximum out-of-plane displacement of all specimens was not greater than 2.5 mm, which is still within the controllable range. The strength enhancement coefficient in Table 3 was inversely proportional to the concrete strength, indicating that the constraint effect of the external steel tube on the core concrete gradually decreased with the increase in concrete strength.

Figure 16 shows the Mises stress cloud maps of components with different concrete strengths. As shown in the figure, both types of specimens exhibited prominent component bending phenomena, and the stress distribution patterns were basically the same. However, the bending phenomenon of the component with a concrete strength of 30 MPa was more obvious. The maximum displacement values of the non-node area steel tube along the *y*-axis of the two types of specimens were 25.166 mm and 15.448 mm, respectively, showing that the non-node area steel tube concrete of the upper and lower limbs protruded in different directions. The overall performance was an S-shaped bending. This was mainly because when the concrete strength was higher, the degree of inward bending of the outer steel tube decreased, indicating that the concrete strength impacts on the degree of bending of the component when it is damaged.

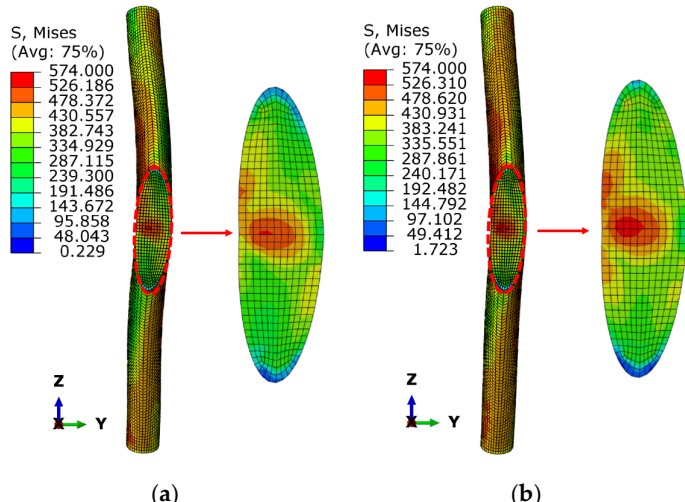

**Figure 16.** Mises stress nephogram of members with different concrete strength. (**a**) $f_c$ = 30 MPa. (**b**) $f_c$ = 120 MPa.

## 7. Dimensionless Skeleton Curve Model

Based on the simulation results of axial cyclic loading at the OMSIN, the key points of the skeleton curve are calculated in this section. Due to the differences in mechanical parameters such as initial stiffness, ultimate bearing capacity, and ultimate deformation in the axial tension and compression directions, skeleton curve models are established for

the OMSIN in the axial compression and tension directions, respectively [35]. Due to the differences in parameter settings in finite element analysis, there is a large dispersion in the ultimate bearing capacity and corresponding displacement under different factor levels. The skeleton curve based on absolute values has a significant difference. Therefore, in this section, the skeleton curve of the OMSIN is simplified into a dimensionless axial compression three-line and axial tension two-line skeleton curve based on peak load and peak displacement to reflect the characteristic points of the nodes.

Figure 17 shows the simplified dimensionless skeleton curve model, where points A, B, and C represent the yield point, peak point, and ultimate point in the axial compression direction, respectively; points A* and B* represent the yield point and peak point in the axial tension direction, respectively; $\delta$ is displacement; $\delta_m$ is the displacement corresponding to the peak load; $F$ is load; and $F_u$ is the peak load. The dimensionless processing is performed on the skeleton curves of 23 intersection nodes, and the coordinates of each characteristic point are obtained, as shown in Table 4.

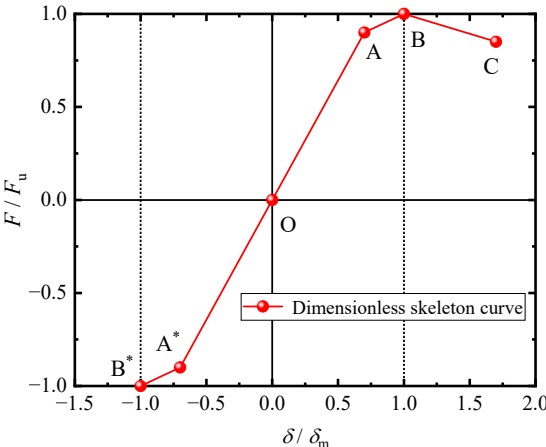

**Figure 17.** Dimensionless Skeleton Curve Model.

**Table 4.** Dimensionless Skeleton Curve Feature Points.

| Name of the Test Piece | Yield Point | | Peak Point | | Limit Point | |
|---|---|---|---|---|---|---|
| | $\delta_y/\delta_m$ | $F_y/F_m$ | $\delta_m/\delta_m$ | $F_m/F_m$ | $\delta_u/\delta_m$ | $F_u/F_m$ |
| Node-Standard | 0.651 (−0.431) | 0.964 (−0.848) | 1.00 (−1.00) | 1.00 (−1.00) | 2.667 | 0.926 |
| Node-Space1 | 0.734 (−0.421) | 0.948 (−0.853) | 1.00 (−1.00) | 1.00 (−1.00) | 2.667 | 0.910 |
| Node-Space2 | 0.704 (−0.430) | 0.945 (−0.854) | 1.00 (−1.00) | 1.00 (−1.00) | 2.667 | 0.900 |
| Node-Space4 | 0.416 (−0.370) | 0.956 (−0.897) | 1.00 (−1.00) | 1.00 (−1.00) | 2.667 | 0.996 |
| Node-Space5 | 1.000 (−0.299) | 0.998 (−0.783) | 1.00 (−1.00) | 1.00 (−1.00) | 4.000 | 0.718 |
| Node-Sym0.70 | 0.682 (−0.444) | 0.926 (−0.830) | 1.00 (−1.00) | 1.00 (−1.00) | 2.667 | 0.882 |
| Node-Sym0.50 | 0.609 (−0.948) | 0.863 (−0.988) | 1.00 (−1.00) | 1.00 (−1.00) | 2.667 | 0.709 |
| Node-Sym0.39 | 0.591 (−0.385) | 0.830 (−0.849) | 1.00 (−1.00) | 1.00 (−1.00) | 2.667 | 0.846 |
| Node-Plane20 | 0.643 (−0.457) | 0.939 (−0.816) | 1.00 (−1.00) | 1.00 (−1.00) | 2.667 | 0.946 |

| Node-Plane50 | 0.674 (−0.410) | 0.956 (−0.856) | 1.00 (−1.00) | 1.00 (−1.00) | 2.667 | 0.873 |
|---|---|---|---|---|---|---|
| Node-Plane70 | 0.647 (−0.426) | 0.915 (−0.837) | 1.00 (−1.00) | 1.00 (−1.00) | 2.667 | 0.854 |
| Node-Plane90 | 0.609 (−0.411) | 0.904 (−0.844) | 1.00 (−1.00) | 1.00 (−1.00) | 2.667 | 0.868 |
| Node-Con0.00 | 1.000 (−0.614) | 0.985 (−0.867) | 1.00 (−1.00) | 1.00 (−1.00) | 4.000 | 0.900 |
| Node-Con0.05 | 1.000 (−0.577) | 0.998 (−0.852) | 1.00 (−1.00) | 1.00 (−1.00) | 4.000 | 0.873 |
| Node-Con0.0.01 | 0.667 (−0.387) | 0.933 (−0.874) | 1.00 (−1.00) | 1.00 (−1.00) | 2.667 | 0.749 |
| Node-Con0.20 | 0.662 (−0.442) | 0.971 (−0.861) | 1.00 (−1.00) | 1.00 (−1.00) | 2.667 | 0.914 |
| Node-Steel0.05 | 0.950 (−0.463) | 0.948 (−0.969) | 1.00 (−1.00) | 1.00 (−1.00) | 4.000 | 0.759 |
| Node-Steel0.20 | 0.803 (−0.473) | 0.938 (−0.900) | 1.00 (−1.00) | 1.00 (−1.00) | 1.333 | 0.891 |
| Node-Steel0.25 | 0.744 (−0.470) | 0.935 (−0.853) | 1.00 (−1.00) | 1.00 (−1.00) | 2.667 | 0.881 |
| Node-Concrete30 | 0.516 (−0.457) | 0.887 (−0.847) | 1.00 (−1.00) | 1.00 (−1.00) | 1.600 | 1.001 |
| Node-Concrete70 | 0.796 (−0.444) | 1.000 (−0.834) | 1.00 (−1.00) | 1.00 (−1.00) | 1.600 | 0.990 |
| Node-Concrete90 | 0.720 (−0.443) | 0.943 (−0.840) | 1.00 (−1.00) | 1.00 (−1.00) | 2.667 | 0.962 |
| Node-Concrete120 | 0.648 (−0.433) | 0.938 (−0.843) | 1.00 (−1.00) | 1.00 (−1.00) | 2.667 | 0.891 |

Note: $\delta_y$ and $F_y$ are the yield displacement and yield load of the node, respectively. 0.651 (−0.431) indicates that the abscissa of the positive yield point is 0.651; the abscissa of the negative yield point is 0.431.

According to the dimensionless coordinates of each feature point in the table, linearly fit the regression equations for each stage, as shown in Equations (8) and (9):

Positive:

$$\frac{F}{F_m} = \begin{cases} 1.28672 \, \delta/\delta_m \, ; \delta/\delta_m \leqslant \delta_y/\delta_m \\ 0.1941 \, \delta/\delta_m + 0.80347; \delta_y/\delta_m < \delta/\delta_m \leqslant 1 \\ -0.06574 \, \delta/\delta_m + 1.06318; \delta/\delta_m > 1 \end{cases} \tag{8}$$

Negative:

$$\frac{F}{F_m} = \begin{cases} 1.738896 \, \delta/\delta_m \, ; \delta/\delta_m \leqslant \delta_y/\delta_m \\ -0.20562 \, \delta/\delta_m - 0.78344; \delta_y/\delta_m < \delta/\delta_m \leqslant 1 \end{cases} \tag{9}$$

### 8. Theoretical Bearing Capacity of OMSIN

Reference [21] provided a formula for calculating the bearing capacity of CFST short columns with steel plates to simulate the effect of elliptical connecting plates in the intersection nodes. This formula underestimates the monotonic axial compression bearing capacity of the intersection nodes. However, from the structural design perspective, this formula is conservative and can be applied to practical engineering. See Equation (10).

$$N_{\mathrm{u}} = \left[ f_{\mathrm{ck}} A_{\mathrm{c}} \left( 1 + \frac{f_{\mathrm{s}} \left( A_{\mathrm{s}} + A_{\mathrm{p}} \right)}{f_{\mathrm{c}} A_{\mathrm{c}}} \right) + \left( 1.85 \frac{t_{\mathrm{s}}}{t_{\mathrm{p}}} + 0.12 \right) f_{\mathrm{s}} A_{\mathrm{p}} \right] \times \left( 8.49 \left( \tan a \right)^3 + 1.49 \right) \tag{10}$$

$$A_{\mathrm{c}} = A_{\mathrm{c1}} + A_{\mathrm{c2}} \tag{11}$$

$$A_{\mathrm{c1}} = r \left( \pi - \arccos x \right); A_{\mathrm{c2}} = x \left( 1 - x^2 \right)^{0.5} \tag{12}$$

$$A_{\mathrm{s}} = L_{\mathrm{s}} t_{\mathrm{s}}; A_{\mathrm{p}} = L_{\mathrm{p}} t_{\mathrm{p}} \tag{13}$$

$$L_{\mathrm{s}} = 2r \left( \pi - \arccos x \right); L_{\mathrm{p}} = 2r \left( 1 - x^2 \right)^{0.5} \tag{14}$$

$$x = \frac{\left( r - L \tan a \right)}{r} \tag{15}$$

where $t_{\mathrm{s}}$ is the sum of the steel tube wall thickness and the lining thickness in the node area, $t_{\mathrm{p}}$ is the thickness of the elliptical connecting plate, $A_{\mathrm{c}}$ is the concrete cross-sectional area, $A_{\mathrm{s}}$ is the outer steel tube cross-sectional area, and $A_{\mathrm{p}}$ is the elliptical connecting plate cross-sectional area. The cross-sectional selection and interpretation are shown in Figure 18. It is worth noting that Figure 18 is based on the symmetry of the OMIN, and 1/4 of the node area was considered. The cross-sectional area of the steel tube, core concrete, and elliptical connecting plate in the node area varies with height [21]. The section in the literature is located at a vertical distance of $D/6$ from the centre of the component. The section in this article is selected based on the region where the maximum Mises stress occurred in the node region of the numerical model.

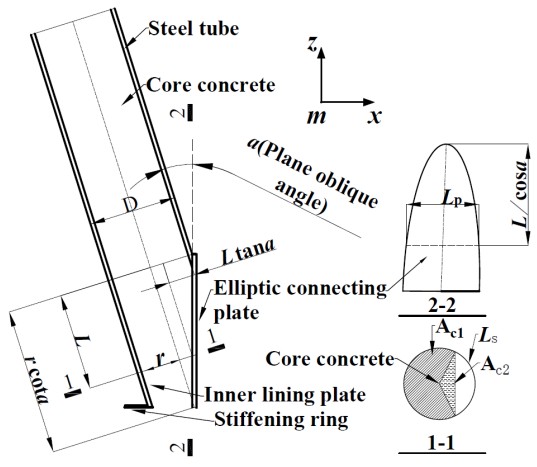

**Figure 18.** Schematic diagram for selecting the cross-section of 1/4 intersecting nodes.

The results calculated using Formula (10) are based on monotonic axial compression loads, and so, they have significant discreteness compared to the ultimate bearing capacity of finite element axial compression. Therefore, multi-parameter optimization was performed on it.

By using SPSS.27 data analysis software and the ultimate bearing capacity and yield load data in Table 3, the formula was fitted and corrected as follows:

$$N_u^{\Theta} = N_u \left[ k_1 \sin \beta + k_2 \cos \alpha + k_3 K + k_4 U + k_5 S + k_6 F + \omega \right] \tag{16}$$

where $\beta$ is the space oblique angle; $\alpha$ is the plane oblique angle; $O$ is the symmetry coefficient; and $S$ is the steel content (%); $K$ is the out-of-plane constraint (kN/mm); $F$ is the compressive strength of concrete (MPa); $k_i$ (i = 1, 2,... 7) is the undetermined coefficient; $\omega$ is a constant term; The undetermined coefficients are shown in Equations (17)–(22):

When calculating the ultimate bearing capacity:

$$k_1 = -1.90156; k_2 = 0.30973; k_3 = 0.14871 \tag{17}$$

$$k_4 = 6.4378 \times 10^{-4}; k_5 = 0.93488; k_6 = -4.08 \times 10^{-3} \tag{18}$$

$$\varpi = 0.86973 \tag{19}$$

When calculating the theoretical yield force:

$$k_1 = -0.96552; k_2 = 0.38402; k_3 = 0.08769 \tag{20}$$

$$k_4 = 5.23 \times 10^{-4}; k_5 = 0.0509; k_6 = -3.52 \times 10^{-3} \tag{21}$$

$$\varpi = 0.7978 \tag{22}$$

Comparing the theoretical calculation values with the ultimate bearing capacity and yield load of the finite element model, as shown in Figure 19. The mean ratio of the two was 0.999 and 1.0323, respectively, and the error was within 20%. Obviously, the two had good consistency and could be used to estimate the node's yield load and ultimate load.

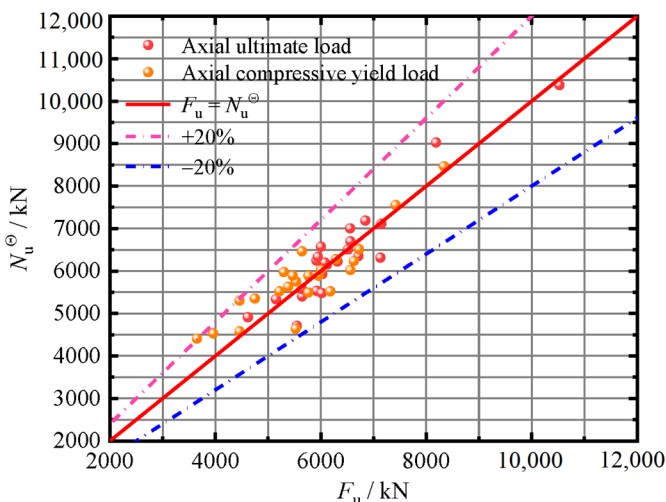

**Figure 19.** Comparison of theoretical calculation values and finite element model values.

## 9. Future Research Prospects

This article conducted numerical analysis and research on the seismic performance of OMSIN under axial reciprocating tension and compression loading, but there were still certain limitations. In future research, the following aspects will be mainly improved: (1) The loading system of this article was symmetrical loading on both sides, but in reality, there was also non-uniformity in the stress on the left and right steel tube concrete columns of the node. In future research, asymmetric seismic loading systems can be considered. (2) Use structural analysis software such as Midas and Sap2000 to conduct an overall analysis of the overall oblique mesh tube. (3) Considering the frequent occurrence of high-rise building fires, in-depth research can be conducted on the mechanical properties after high temperatures and impact resistance of such components. (4) Recently, researchers proposed more novel numerical methods to analyze the stress–strain behaviour of isotropic and anisotropic media. The "Differential Quadrature" method in reference [36] and the "Bezier" method in reference [37] were proven to have higher stability and accuracy than other numerical methods, providing clear improvement ideas for future numerical simulations of OMSIN.

## 10. Conclusions

In this paper, the seismic performance of OMSIN under axial cyclic tension-compression loading was studied using ABAQUS.2022 finite element software. Six parameters were considered: space intersecting angle, symmetry coefficient, plane intersecting angle, out-of-plane restraint, steel content, and concrete strength. Furthermore, a dimensionless skeleton curve model of the OMSIN was established. Finally, a multi-parameter fitting correction was performed on the formula for the ultimate bearing capacity of CFST, and the theoretical calculation formulas for the yield load and ultimate load under axial compression were obtained under cyclic loading of the intersection nodes. The main conclusions are as follows:

1. The ABAQUS three-dimensional solid model based on the core concrete constitutive model and the five-stage quadratic flow model used in this paper can effectively reflect the interaction between components of the OMSIN in seismic cyclic loading and can provide theoretical guidance for subsequent related research.

2. The increase in space angle significantly weakens effect on the seismic performance, and the space angle directly affects the failure mode of the nodes. Components with a space angle ≥ 4°are difficult to control regarding out-of-plane displacement, which can easily cause lateral instability failure. Moreover, large out-of-plane deformation has already occurred when the axial displacement is small. Therefore, it is not recommended to use such components in engineering practice due to economic and safety considerations. The asymmetric arrangement of the plane angle will cause the nonlinear development of out-of-plane displacement. When the symmetry coefficient is less than 0.70, the reduction in the ultimate bearing capacity of the nodes is significant. Larger out-of-plane and vertical deformation restraints should be applied for nodes with inconsistent upper and lower limb plane angles.

3. The specimens' ultimate load and overall compressive stiffness are positively correlated with the plane angle. Vertical restraints should be applied to the node positions of components with a plane angle ≥ 70° to reduce vertical residual deformation. Out-of-plane restraint is a key factor affecting the seismic performance of the nodes, and it is proportional to the ultimate load of the components. However, when the ratio of the diameter of the steel rod to the outer diameter of the steel pipe used for out-of-plane constraints is ≥0.15, the change in out-of-plane constraints has no significant improvement effect on the seismic performance of the component.

4. The steel content is positively correlated with the components' positive and negative ultimate loads and overall stiffness. In structural design, if the aim is to improve the mechanical performance of the components by increasing the steel content, larger

out-of-plane restraints should be set to utilize the material properties fully. The concrete strength is proportional to the positive ultimate load but has little effect on the negative ultimate load and hysteresis curve evolution. Components with higher concrete strength have a faster decay rate of axial compression stiffness due to their high brittleness in the later stages. When increasing the concrete strength, the hoop coefficient of the components should be increased simultaneously.

5. Three-line and two-line models were selected for the axial compression direction, and axial tension direction, respectively. A dimensionless skeleton curve model based on peak load and peak displacement of OMSIN was established. By fitting and modifying the formula for the ultimate bearing capacity of CFST, a formula for calculating the axial compressive yield load and ultimate load that can reflect the influence of all parameters is obtained. The theoretical values are in good agreement with the calculated values of the finite element model and can be used for estimating the mechanical characteristic points of OMSIN under the influence of multiple parameters.

**Author Contributions:** Conceptualization, J.Z. and B.Y.; methodology, F.W.; software, F.W.; validation, J.Z. and B.Y.; formal analysis, F.W.; investigation, F.W.; resources, B.M.; data curation, B.M.; writing—original draft preparation, F.W.; writing—review and editing, F.W. and B.Y.; visualization, B.Y.; supervision, B.M.; project administration, J.Z.; funding acquisition, J.Z. and B.M. All authors have read and agreed to the published version of the manuscript.

**Funding:** This research was supported by the National Natural Science Foundation of China, grant number (12162010). Guangxi Science and Technology Plan Project, grant number (Guike AD20159085, Guike AA20302006).

**Institutional Review Board Statement:** Not applicable.

**Informed Consent Statement:** Not applicable.

**Data Availability Statement:** The data used to support the findings of this study are available from the corresponding author upon request.

**Conflicts of Interest:** The authors declare no conflict of interest.

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
