# Peer review of "Seismic Behaviour of CFST Space Intersecting Nodes in an Oblique Mesh"

_applsci, doi:10.3390/app13105943_

Round 1

Reviewer 1 Report

Keywords: the keywords “oblique mesh” start in lowercase while the other keywords start with an uppercase letter. Either start all keywords with an uppercase letter or start all keywords with a lowercase letter (except cases where the word always starts with an uppercase letter).

References: there are references not mentioned/used in the article text (like 2, 3, 5, 6, 7, 8, etc.). References have to be mentioned or cited in the article text. Either remove unused references or use them in the article text.

Figure 2: What program was used to make these 3D models? What techniques and methods were used in their creation?

Formulas 1-7: What are each formula’s relation to one another? Why were these specific 7 formulas used and what were they used for? What are the obtained results from these formulas?

Figure 4, 6, 8, 10, 12, 14 and 16: Same notes as Figure 2: what programs were used for 3D models and what were the techniques/methods used?

A “Results” section is missing right before the “Conclusion” section.

There seem to be two “6” chapters: “6. Theoretical bearing capacity of OMSIN” and “6. Conclusion”. Change the chapter numbers to be in correct order.

Line 674: “Data Availability” is not bolded fully, only the first letter is.

Acknowledgments: The parts about funding should be mentioned in a separate “Funding” section instead.

The end of the article is missing certain sections mentioned in the MDPI template, such as “Author contributions”, “Funding”, etc.

References: References should not be numbered using “[1]” format, and instead should use a more standard format, like “1.”.

The node titles in Tables (ex. "Node-3-35-0.15-0.13- 90") are very complicated and easy to mix up. Consider shortening/renaming them into more easy to understand names.

Reviewer 2 Report

This study investigates the mechanical performance of space-intersecting nodes of oblique meshes under cyclic axial tension and compression loads using numerical analysis. Six parameters were considered, and the study found that changes in steel tube wall thickness were unfavorable for stress transmission. Increasing the space intersecting angle weakened seismic performance, and the ultimate load and overall compressive stiffness were positively correlated with the plane angle. The study also established a dimensionless skeleton curve model for the node and a calculation formula for the axial yield and ultimate load under cyclic loads. However, before further consideration of your manuscript, you must fully “address” the comments listed below:

1-    ABSTRACT would benefit from a more structured approach. Commence by presenting a comprehensive summary of the topic along with the rationale behind the paper. Proceed to outline the methodology employed and the manuscript's general structure. Additionally, elaborate further on the outcomes by providing numerical details towards the conclusion.

2-     How does the steel content affect the failure mode of space-intersecting nodes under cyclic loads? 

3-    What is the effect of the symmetry coefficient on the out-of-plane displacement of space-intersecting nodes?

4-    How can the results of this study be used to enhance the safety and resilience of structures in seismic regions?

5- The novelty of your work requires further elucidation to the reader, necessitating additional detail in both the Abstract and Introduction. In essence, the research's purpose is not adequately conveyed and therefore should be clearly presented.

6-    It is still not clear how equations 1 to 7 (stress-strain relationship of steel tube-confined core concrete) are derived.

7-    XYZ coordinates and scale bars are missing from many figures. 

8- Kindly provide a comprehensive introduction of the elastic properties of all structural components expounded upon in your manuscript. It would be appropriate to compile them into a summary table.

9-    Provide more explanation for this sentence: Page 12, Line 342: " However, the degradation rate of the later stiffness increases, mainly due to the asymmetric arrangement, which causes the lower limb steel tube concrete to be in a complex stress state of compression, bending, and shear”.

10-  Similarly, this sentence needs to be better explained: Page 13, Line 353: “However, this component also exhibits a trend of nonlinear growth in out-of-plane displacement”.

11- Apart from the employed numerical simulations (Abaqus software), more novel and strong numerical methods have been recently proposed for the stress-strain analysis of isotropic and anisotropic media. Among them, the “Differential Quadrature” and “Bezier” methods proved to have higher stability and accuracy than other numerical methods. For this purpose, please write a paragraph in your paper introducing these methods which can “alternatively” investigate the mechanical performance of space intersecting nodes of oblique meshes under cyclic axial tension and compression loads  (and reference the papers listed below):

“Differential Quadrature Method”:

·      Khalid, H. M., Ojo, S. O., & Weaver, P. M. (2022). Inverse differential quadrature method for structural analysis of composite plates. Computers & Structures, 263, 106745.

“Bezier Method”:

·      Kabir, H., & Aghdam, M. M. (2021). A generalized 2D Bézier-based solution for stress analysis of notched epoxy resin plates reinforced with graphene nanoplatelets. Thin-Walled Structures169, 108484.

12- Conclusion: Can authors highlight future research directions and recommendations? Also, highlight the assumptions and limitations (e.g., shortcomings of the present study). Besides, recheck your manuscript and polish it for grammatical mistakes (you can use “Grammarly” or similar software to quickly edit your document).

Extensive editing of English language is required

Round 2

Reviewer 2 Report

The authors fully addressed my comments. 

English is decent in the current format.